# Electrothermal mineralization of per- and polyfluoroalkyl substances for soil remediation

Yi Cheng [1,15], Bing Deng [1,14,15] ✉, Phelecia Scotland[1,2], Lucas Eddy[1,3,4], Arman Hassan[5], Bo Wang[6,7], Karla J. Silva[1], Bowen Li[1], Kevin M. Wyss[1], Mine G. Ucak-Astarlioglu [8], Jinhang Chen [1], Qiming Liu [1], Tengda Si[1], Shichen Xu[1], Xiaodong Gao[9,10], Khalil JeBailey [2], Debadrita Jana[9], Mark Albert Torres[9], Michael S. Wong [1,6,7,11], Boris I. Yakobson [1,2,4], Christopher Griggs[8], Matthew A. McCary[5], Yufeng Zhao [2,12] ✉ & James M. Tour [1,2,4,13] ✉

Per- and polyfluoroalkyl substances (PFAS) are persistent and bioaccumulative pollutants that can easily accumulate in soil, posing a threat to environment and human health. Current PFAS degradation processes often suffer from low efficiency, high energy and water consumption, or lack of generality. Here, we develop a rapid electrothermal mineralization (REM) process to remediate PFAS-contaminated soil. With environmentally compatible biochar as the conductive additive, the soil temperature increases to >1000 °C within seconds by current pulse input, converting PFAS to calcium fluoride with inherent calcium compounds in soil. This process is applicable for remediating various PFAS contaminants in soil, with high removal efficiencies ( >99%) and mineralization ratios ( >90%). While retaining soil particle size, composition, water infiltration rate, and cation exchange capacity, REM facilitates an increase of exchangeable nutrient supply and arthropod survival in soil, rendering it superior to the time-consuming calcination approach that severely degrades soil properties. REM is scaled up to remediate soil at two kilograms per batch and promising for large-scale, on-site soil remediation. Life-cycle assessment and techno-economic analysis demonstrate REM as an environmentally friendly and economic process, with a significant reduction of energy consumption, greenhouse gas emission, water consumption, and operation cost, when compared to existing soil remediation practices.

Per- and polyfluoroalkyl substances (PFAS) are a diverse class of anthropogenic chemicals that are extensively used in plastics, textiles, food wrapping materials, and fire-fighting foams[1,2]. PFAS can easily accumulate in soil through waste disposal and animal migration and has been proven to be bioaccumulative and toxic to humans and wildlife[3–6]. Due to the high bond energy of C-F (~485 kJ mol⁻¹)[7] and resulting long half-lives (>100 years in soils)[8], the efficient elimination of PFAS is difficult to realize by natural decomposition or microbiological treatment[8–10].

Many efforts have been devoted to the remediation of PFAS-contaminated soil in the past decade, mainly including stabilization[11–13], chemical oxidation[14–16], and thermal treatment[17–19]. The stabilization

method involves mixing sorbents, such as activated carbon or clay, with the contaminated soil to sorb PFAS and reduces PFAS mobility and bioavailability[11–13]. However, this method does not degrade PFAS in soil and sorbed PFAS could still pose long-term environmental damage. Chemical treatment uses strong oxidants to oxidize PFAS[14–16]. The residual oxidants need to be washed out with a large amount of water to avoid its damage to the soil, where the wastewater could lead to secondary pollution to the environment. Traditional thermal treatment requires furnace heating for PFAS desorption and degradation, which often lasts for hours at 400-1100 °C[17–20]. Some toxic short-chain fluorocarbon compounds, such as $CF_4$, $C_2F_6$, and $C_2F_4$, could be generated and emitted to the environment during this process. This is due to inadequate decomposition of C-F bonds, which will cause secondary pollution[20,21], and the extended heating also degrades soil properties[22].

Converting PFAS into non-toxic metal fluoride with the aid of alkali or alkaline earth metal ions like calcium ion ($Ca^{2+}$) under thermal treatment, termed as mineralization, is promising for PFAS degradation[23–26]. However, traditional furnace heating often lasts hours, consuming large amounts of energy and the PFAS mineralization ratios are typically <80%. More importantly, additional calcium compounds are always required for PFAS mineralization, leading to the high materials consumption[23–26]. Hence, developing an efficient, economical and general thermal process for remediation of PFAS-contaminated soil is highly desirable, especially if the soil can remain in place and need not be excavated and transported. The emerging direct electric heating techniques, possessing the merits of rapid heating and cooling rates, short treatment duration and thus ultralow energy consumption[27–35], can provide a promising opportunity for PFAS mineralization.

Here, we developed a rapid electrothermal mineralization (REM) method for the effective remediation of PFAS-contaminated soil. Using environmentally compatible biochar as the conductive additive, the temperature of contaminated soil rapidly escalates to >1000 °C within seconds through a direct current pulse input, with an ultrafast heating ($\sim$$10^4$ °C s$^{-1}$) and cooling rate ($\sim$$10^3$ °C s$^{-1}$). During REM, by virtue of the high Ca content inherent in soil and biochar, PFAS can be mineralized into calcium fluoride ($CaF_2$), a natural occurring and non-toxic mineral. This REM process conducted in a sealed system produces negligible harmful fluorocarbon gas emissions. High removal efficiencies (>99%) and fluorine mineralization ratios (>90%) for various PFAS were simultaneously realized, demonstrating the broad applicability of the REM process. REM facilitates an increased exchangeable nutrient supply of the treated soil, while maintaining soil particle size, composition and water infiltration rate, rendering it superior to the time-extended calcination approach that severely degrades soil properties. When further used for arthropod culture, REM soil exhibits a comparable arthropod survival ratio with the clean raw soil, while arthropods die rapidly in the PFAS-contaminated soil. Remediation of soil on the kilogram scale per batch has been accomplished here, suggesting the potential applicability of REM for large-scale deployment. Furthermore, life-cycle assessment shows that REM exhibits low energy-consumption ($\sim$420 kWh ton$^{-1}$), no water consumption, and minimal greenhouse gas emission, making it an environmentally attractive alternative over existing remediation techniques.

## Results and discussion
### REM for the remediation of PFOA-contaminated soil
A conceptional design of on-site REM is shown in Fig. 1a, which leverages mature agricultural techniques and soil remediation practices. In the first step, contaminated soil is premixed with conductive additives, such as biochar, to ensure appropriate electrical conductivity. In the second step, the electrodes fixed in an insulating cap are inserted into the soil. A high-voltage pulse input within seconds controllably brings the soil to a typical temperature of

>1000 °C, facilitating the rapid mineralization of toxic PFAS, with existing Ca compounds in soil and biochar into the nontoxic natural mineral, $CaF_2$.

We initially performed a proof-of-concept test of the REM process on a bench scale (Fig. 1b and Supplementary Fig. 1). Raw soil was collected from the Rice University campus (see Methods for details), which contains undetectable content of PFAS (<1 ppb) by liquid chromatography-mass spectrometry (LC-MS). The raw soil was separately spiked with different kinds of PFAS with the content of ~100 ppm (Supplementary Table 1). PFAS-contaminated soil was mixed with appropriate amounts of biochar, and then loaded into a quartz tube reactor. No additional Ca-containing compound was added, considering there are sufficient Ca species inherent in the soil for PFAS mineralization. The sample resistance was regulated by compressing the graphite electrodes inserted at the end of the quartz tube, which were connected to a capacitor bank. In a typical experiment, with an input voltage of 100 V, discharging time of 1 s, and sample resistance of 3.5 Ω (Supplementary Table 2), the peak current reaches ~140 A (Fig. 1c) and the peak temperature is ~1370 °C (Fig. 1d). The heating and cooling rates during REM were calculated to be ~$10^4$ °C s$^{-1}$ and ~$10^3$ °C s$^{-1}$, respectively, using an infrared thermometer. By tailoring the input voltage from 40 to 150 V, the REM temperature is tunable ranging from 300 to 2500 °C (Supplementary Figs. 2 and 3), which meets the required temperature of PFAS degradation, as determined by thermogravimetric analysis (TGA, Supplementary Fig. 4). After REM, the residual PFAS content was quantified by high-performance liquid chromatography with a diode array detector (HPLC-DAD, Supplementary Fig. 5) and triple quadrupole LC-MS system (QQQ LC-MS, Supplementary Fig. 6). The detecting limits of each PFAS characterization methods are listed in Supplementary Table 3. The mineralized fluorine ion (F$^-$) content was tested by ion chromatography (IC, Supplementary Fig. 7).

We first investigated the degradation process of perfluorooctanoic acid (PFOA), a representative type of PFAS. REM was initially conducted in an open system without O-rings to seal the quartz tube. With the increase of input voltage, the PFOA content decreases, benefitting from a higher reaction temperature (Supplementary Fig. 8). However, the total fluorine content significantly decreases with the increase of input voltage, with only half of the organic fluorine mineralized into fluorine ions, which can be ascribed to the emission of PFOA-degraded short-chain species (Supplementary Fig. 9). To avoid the emission of short-chain fluorocarbon species, we constructed a sealed reactor with two O-rings on each electrode to seal the reactor tube during REM (Fig. 1b and Supplementary Fig. 1c). With the increase of input voltage, the PFOA content progressively decreased, benefiting from a higher reacting temperature. Consequently, the F$^-$ content increased with an increase of input voltage from 0 to 100 V and an optimal mineralization ratio of 94% was obtained (Fig. 1e). By virtue of the sealing design, REM soil shows a much higher mineralization ratio (94%) compared with the furnace-calcined soil (0.34%), while keeping a high and comparable PFOA removal efficiency of >99% (Supplementary Fig. 10). The gaseous byproduct was collected and tested by gas chromatography mass spectrometry (GC-MS, Supplementary Fig. 11). Compared to the clean raw soil as the control, no additional peaks corresponding to known PFOA degradation products were observed (Supplementary Fig. 12). On the contrary, some PFOA-degraded fluorinated compounds, such as $CF_4$, $CH_3F$, $C_2F_6$, $C_2F_4$, and $C_6H_5F$, were observed when replacing soil with $SiO_2$ (Supplementary Fig. 13). This indicates that REM in the presence of Ca effectively mineralizes F from soil contaminated with PFAS and avoids the emission of PFAS degraded short-chain fluorocarbon species. Thus, the total fluorine mass was calculated by adding the organic fluorine in residual PFOA and the mineralized F$^-$ (Fig. 1e). The slight decrease in mineralization ratio and quantifiable total fluorine mass when the

REM voltage increases from 100 V to 150 V (Fig. 1e), may be attributed to the increased amount of insoluble F-containing compounds deposited on the quartz tube with the increase of REM temperature (Supplementary Fig. 14).

The PFOA content in the soil can be reduced to below the residential soil remediation standards (130 ppb, the New Jersey Department of Environmental Protection, ref. 36), after 2 electric pulses and further to an ultralow value of ~1.1 ppb after 4 electric pulses (Fig. 1f). [19]F NMR spectra were conducted using deuteroxide to extract PFOA and F⁻ in the soil before and after REM (Fig. 1g). The [19]F NMR spectrum of contaminated soil has several peaks, all of which fit well with PFOA standard[37,38]. On the contrary, REM-treated soil has a single peak at -128 ppm, corresponding to hydrated fluoride ions[37], further proving the PFOA can be effectively converted into fluorine ions in the soil by the REM process.

## Generality of REM for soil remediation

To demonstrate the generality of REM for PFAS degradation, other than PFOA, we investigated the mineralization behaviors of various PFAS, including heptadecafluorooctanesulfonic acid tetra-ethylammonium salt (PFOS), tridecafluorohexane-1-sulfonic acid potassium salt (PFH$_x$S), and nonafluorobutane-1-sulfonic acid potassium salt (PFBS). The trends of PFAS mineralization versus input voltage are similar to that of PFOA, where higher input voltages often facilitate higher degradation ratios of C-F bonds (Fig. 2a–c). In the [19]F NMR spectra, only the −128 ppm peak that assigned to hydrated F⁻ (ref. 37) was observed for all REM-treated soil samples (Fig. 2d–f), indicating the effective removal of PFAS by REM. The removal efficiencies of all the tested PFAS were calculated to be >99% (Fig. 2g) and >90% mineralized fluorine ratios were quantified with a single electric pulse (Fig. 2h). In addition to short-chain PFAS, REM is also applicable

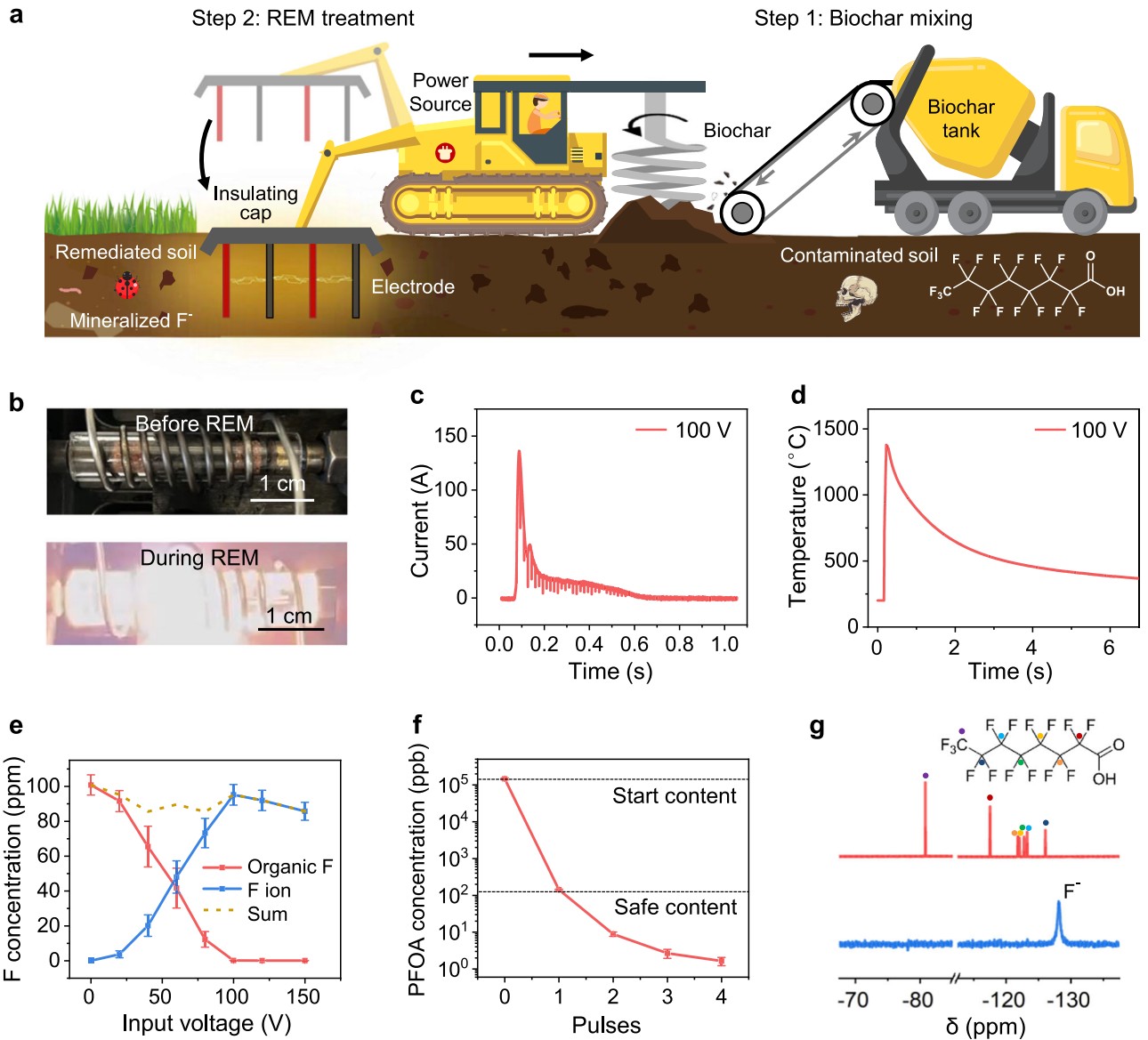

**Fig. 1 | Rapid electrothermal mineralization (REM) process for the remediation of PFOA-contaminated soil. a** Conceptional schematic of REM process for bulk soil remediation. **b** Pictures of the sample before (top) and during (bottom) the REM reaction. A spring coiled around the quartz tube is used to increase the mechanical integrity of the tube. **c** Current curve with the input voltage of 100 V and duration time of 1 s. **d** Real-time temperature curve at an electric input of 100 V for 1 s recorded by an infrared thermometer. The temperature detection range of the thermometer is 200−1500 °C. **e** Concentrations of organic fluorine (red line) and mineralized fluorine ion (blue line) in PFOA-contaminated soil varied with input voltages. **f** Residual PFOA concentrations in soil after repetitive electric pulses, with voltage of 100 V and duration of 1 s each time. The error bars in **e** and **f** denote standard deviations, where $N = 3$. **g** [19]F NMR spectra of the PFOA-contaminated soil extractant before (top) and after (bottom) REM. Inset, the molecular structure of PFOA.

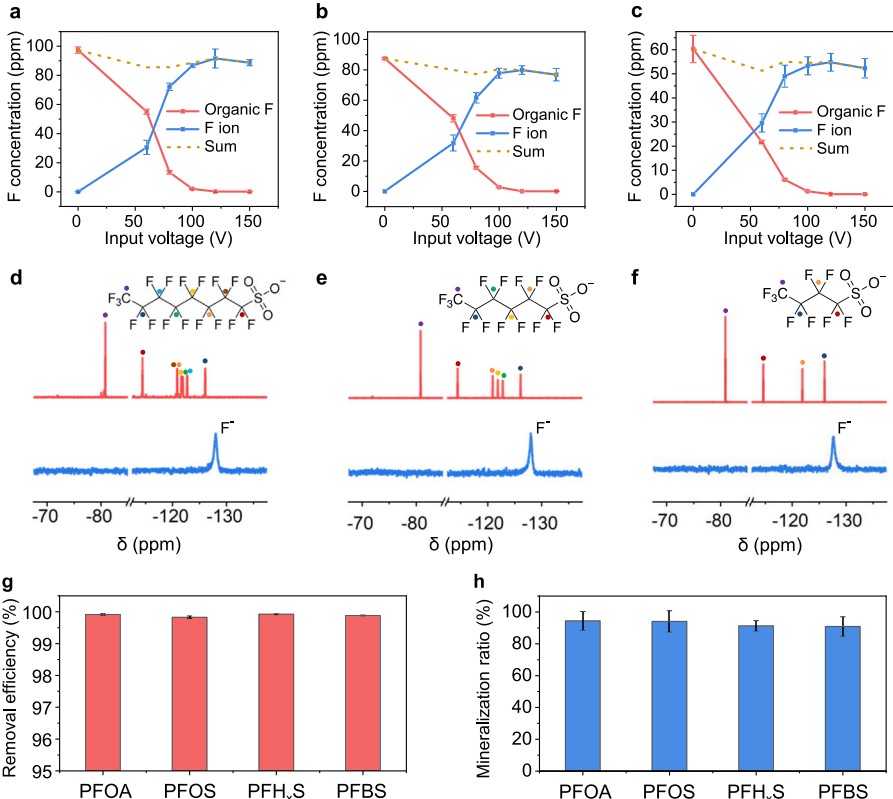

**Fig. 2 | Generality of REM process for various PFAS. a–c** Concentrations of organic fluorine (red line) and mineralized fluorine ion (blue line) varied with input voltages for **a** PFOS-contaminated soil, **b** PFHₓS-contaminated soil, and **c** PFBS-contaminated soil. **d–f** ¹⁹F NMR spectra of the **d** PFOS, **e** PFHₓS, and **f** PFBS -contaminated soil extractant before (top) and after (bottom) REM. Insets in **d–f** are the molecular structure of PFOS, PFHₓS, and PFBS, respectively. The dots denote the F peaks and corresponding F attached C atoms. **g** Removal efficiencies of different kinds of PFAS. **h** Mineralization ratios of different kinds of PFAS. The error bars in **a–c**, **g**, and **h** denote standard deviations, where $N = 3$.

to mineralize F-containing polymers, such as polytetrafluoroethylene (PTFE) with a high mineralization ratio of ~95% (Supplementary Fig. 15). Trace amounts of PTFE degradation compounds, including tetrafluoroethylene and trifluoromethanol, were detected in the gaseous phase during REM (Supplementary Fig. 16), while none of the fluorinated compounds were detected in the REM soil (Supplementary Fig. 17).

In addition to biochar, other carbon materials with sufficient conductivity, including carbon black, metallurgical coke (metcoke) and flash graphene[27], were also used as the conductive additives for the REM process. Taking PFOA as an example, all tested carbon conductive additives can achieve a high mineralization ratio of >90% (Supplementary Fig. 18), proving the broad applicability of carbon additives. The used carbon additive can be optionally separated from the soil mixture and then reused for next-batch soil remediation. For example, biochar was separated from soil by dispersion and centrifugation with a recycling recovery of ~85 wt% (Supplementary Figs. 19–21), and reused in a second REM process with a comparable PFAS mineralization performance (Supplementary Fig. 22). Similarly, when metcoke was used as the conductive additives, ~91 wt% can be recycled after REM by simply sieving (Supplementary Figs. 23 and 24) and then reused with similar performance (Supplementary Fig. 25). This significantly reduces materials consumption of REM while requiring greater processing. The optimal ratio between soil and different carbon additives was also investigated, where sufficient carbon additive content (>33 wt%) is required to ensure REM temperature for PFAS mineralization (Supplementary Fig. 26). For deployed applications, the choice of carbon additives depends on the specific scenarios and requirements.

## Mechanism of PFAS mineralization

$Ca^{2+}$ is suggested to be a critical counterion for PFAS mineralization under thermal treatment[24,25]. To confirm the influence of Ca on PFAS mineralization, we first compared the mineralization performance of $Ca^{2+}$ with other alkali and alkaline earth metal ions, such as $Mg^{2+}$ and $Na^+$, where calcium carbonate ($CaCO_3$, a representative calcium specie in soil[39]), magnesium carbonate ($MgCO_3$) and sodium carbonate ($Na_2CO_3$), were separately mixed with PFOA and the metal counterion content is 1.2 mole equivalent compared with F (Supplementary Table 4). After REM treatment, X-ray diffraction (XRD) patterns show the loss of PFOA peaks and the appearance of metal fluoride peaks (Supplementary Figs. 27 and 28), indicating that all these alkaline and alkaline earth metal ions can be used for PFAS mineralization. However, Ca achieved the highest PFOA removal efficiency (~99.7%) over Mg (~94.2%) and Na (~98.5%, Supplementary Fig. 29), proving that Ca has the best mineralization performance for PFAS, greater than Mg and Na. Meanwhile, the Ca-F bond has the highest bond energy among different metal-fluorine bonds (Supplementary Table 5). Theoretical calculation reveals that once other metal fluorides (like $MgF_2$ or NaF) are formed, these fluoride compounds are thermodynamically favorable to convert to $CaF_2$ during REM at temperatures higher than 400 °C (Supplementary Fig. 30). The above analysis evinces that Ca dominates the PFAS mineralization process. Then, we mixed $CaCO_3$ with other kinds of PFAS and observed the same mineralization phenomena (Supplementary Figs. 27 and 31), explicitly demonstrating the critical role of $Ca^{2+}$ in PFAS mineralization.

When biochar was used as the conductive additive, a slightly higher mineralization ratio of ~94% was observed, compared to that of other carbon additives (90–91%, Supplementary Fig. 16). We examined

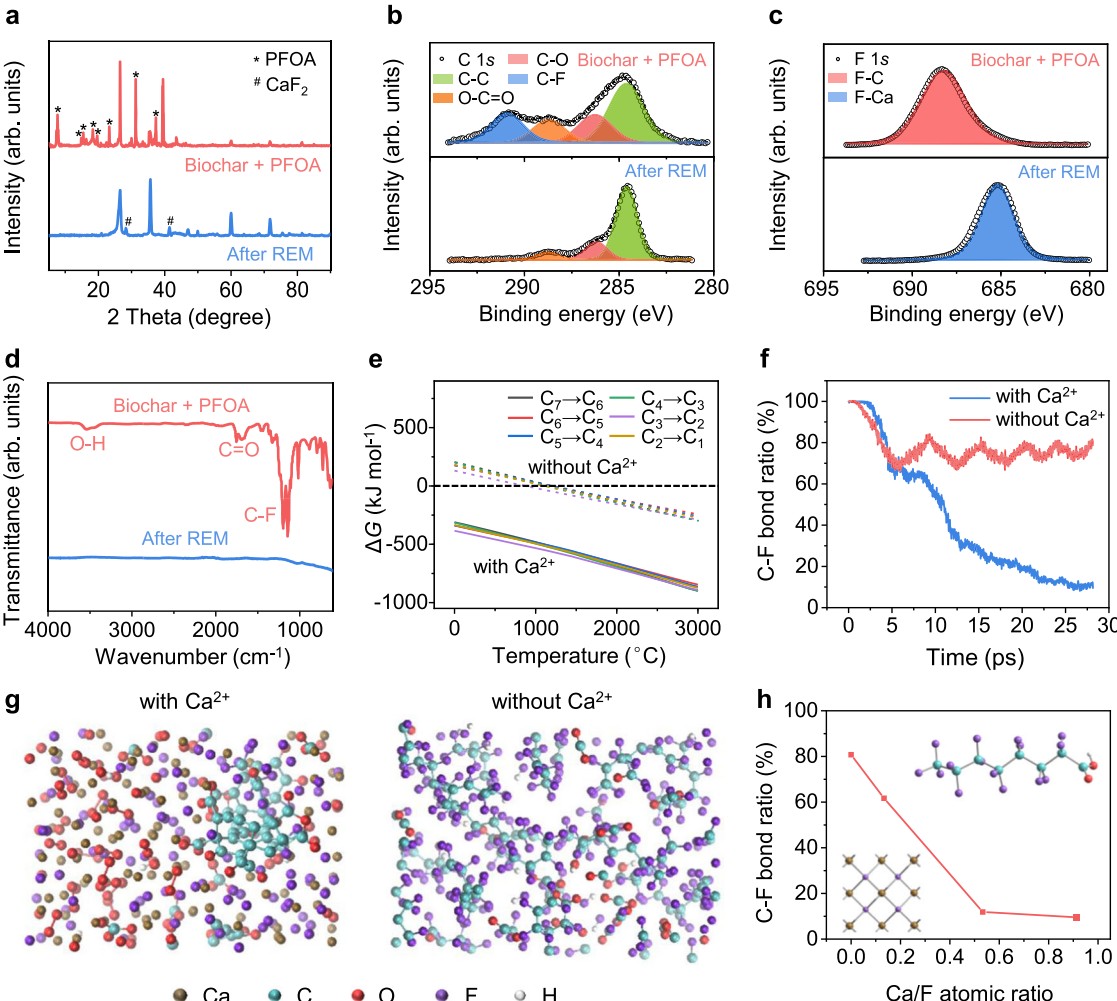

**Fig. 3 | Mechanism of PFAS mineralization during the REM process. a** XRD patterns of PFOA/biochar before (red) and after (blue) REM. **b** C 1s XPS fine spectra of PFOA/biochar before (top) and after (bottom) REM. **c** F 1s XPS fine spectra of PFOA/biochar before (top) and after (bottom) REM. **d** IR spectra of PFOA/biochar before (red) and after (blue) REM. **e** Gibbs free energy change (ΔG) of each PFOA degradation step with (solid line) and without (dash line) Ca²⁺ varied with temperature. The black dash line denotes ΔG = 0 kJ mol⁻¹. **f** Simulated variation of C-F bond ratio during REM process with calcium (blue line, F105Ca96) and without calcium (red line). **g** Optimized structure snapshots after simulated heating treatment with calcium (left, F105Ca96) or without calcium (right). **h** Relationship between C-F bond ratio after REM and the input atomic ratio of calcium and fluorine. Insets are the structures of PFOA (top right) and mineralized CaF₂ (bottom left).

the composition of these carbon additives by XPS, and found that Ca content is highest in biochar (~4 at%, Supplementary Figs. 32 and 33), while undetectable by XPS in the other carbon additives (Supplementary Fig. 34).

To verify that the Ca²⁺ in biochar can benefit PFAS mineralization, we mixed PFOA (5 wt%) and biochar (95 wt%) and conducted REM. After the treatment, the peaks of PFOA diminished while CaF₂ peak appeared in the XRD patterns (Fig. 3a). The same phenomenon pertains to other PFAS (Supplementary Table 1 and Supplementary Fig. 35). The X-ray photoelectron spectroscopy (XPS) spectra show that the C 1s peak at ~292 eV (assigned to C-F) and the F 1s peak at ~689 eV (assigned to F-C) of PFOA disappeared after REM, while the new F 1s at ~684 eV (assigned to F-Ca) appeared (Fig. 3b, c). In the infrared (IR) spectrum of initial PFOA, the peaks in the range of 1100–1200 cm⁻¹ and 650–750 cm⁻¹ correspond to stretching and rocking vibrations of C-F bonds[40], respectively (Fig. 3d), disappeared after REM. The IR spectra for other PFAS exhibited the same behaviors (Supplementary Fig. 36), demonstrating that Ca in biochar facilitates effective mineralization of PFAS.

Theoretical analysis was further conducted to clarify the PFAS mineralization mechanism assisted by Ca²⁺. Thermodynamically, we calculated the Gibbs free energy change (ΔG) for each degradation step of C₇F₁₆, which is the first-step degradation product of PFOA after decarboxylation[38]. Without Ca²⁺, the thermal pyrolysis of the perfluorinated species requires a high temperature >1500 °C (Fig. 3e, dashed line). In contrast, ΔG turns negative with the existence of Ca²⁺ under a broad temperature range (Fig. 3e, solid line), indicating that the PFOA mineralization reaction is spontaneous. We further performed density functional theory (DFT) and molecular dynamic (MD) simulations to reveal the kinetics of PFOA mineralization. Since the cleavage of C-F bonds is an essential step for PFOA mineralization, the C-F bond ratio is used as an indicator to evaluate the mineralization efficiency. With Ca, F is more favorable to ionically bond with the Ca atom than forming a covalent bond with the C atom. The reaction barrier of C-F bond cleavage is calculated to be 0.67 eV and the total energy is lowered by 1.24 eV in the presence of Ca, indicating that the mineralization process is an energy-favorable reaction step. On the contrary, without Ca, the F atom spontaneously returned to its original

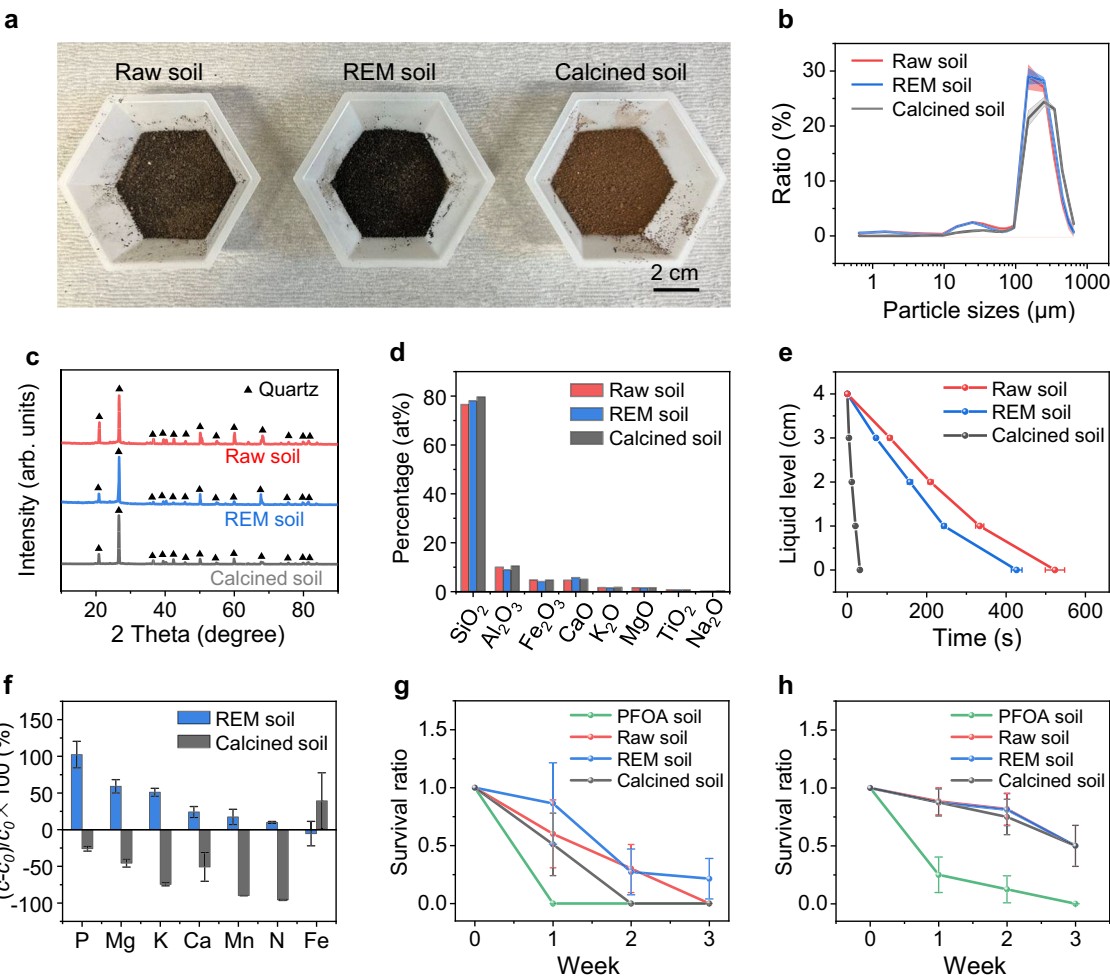

**Fig. 4 | Soil properties after REM treatment. a** Pictures of raw soil (left), REM soil (middle), and calcined soil (right). **b** Particle size distribution of raw soil (red), REM soil (blue), and calcined soil (black). The shadows denote standard deviations, where $N = 5$. **c** XRD patterns of raw soil (red), REM soil (blue), and calcined soil (black). The powder diffraction file (PDF) reference card for quartz, 01-086-1629 (triangle). **d** XRF of raw soil (red), REM soil (blue), and calcined soil (black). **e** Water penetration liquid level varied with time for raw soil (red), REM soil (blue), and calcined soil (black). **f** Exchangeable soil nutrient content change after REM and

calcination processes. $c_0$ and $c$ are the concentrations of exchangeable nutrients in raw soil (blue) and REM soil (black), respectively. The error bars in **e** and **f** denote standard deviations, where $N = 3$. **g** Survival ratio of springtail cultured in different soil samples. **h** Survival ratio of isopod cultured in different soil samples. The error bars in **g** and **h** denote standard deviations, which are calculated from model-predicted values from the generalized linear models with $N = 7$ and $N = 8$, respectively.

position in the PFOA and reformed a covalent bond with the C atom (Supplementary Fig. 37).

Without Ca, ~80% of C-F bonds in PFOA are maintained after thermal treatment in the temperature range of 1500 to 2500 K, and PFOA tends to degrade into short-chain perfluorinated species (Fig. 3f,g). In contrast, with the presence of $Ca^{2+}$, >90% of C-F bonds in PFOA cleave and the F are affiliated to Ca (Fig. 3f, g). With the increase of the atomic ratio of Ca and F, more C-F bonds tends to cleave (Fig. 3h and Supplementary Figs. 38 and 39), demonstrating that more $Ca^{2+}$ can facilitate a higher mineralization ratio of PFAS. The $Ca^{2+}$ content in both soil and biochar, as tested by X-ray fluorescence (XRF) and XPS, was in the range of 4 to 5 at%, which is hundreds of times excess relative to the reaction stoichiometry (Supplementary Figs. 32, 40, and 41). PFAS can thus be effectively mineralized using the inherent $Ca^{2+}$ in soil and biochar, without additional $Ca^{2+}$ consumption, further reducing the materials expense of the REM process.

## Soil properties after REM

The soil properties after REM were investigated, which are significant for the soil reuse in the ecosystem. We compared the soil after REM treatment and carbon additive removal (denoted as REM soil) with raw

soil and calcined soil as a control, since calcination has been reported to be an effective method to remove PFAS from the contaminated soil[17,18].

We first examined the soil physical properties. REM soil exhibits a darker contrast than raw soil, due to the trace residual biochar (Fig. 4a). The calcined soil shows a brick-red contrast, indicating possible composition or structure change during the calcination process (Fig. 4a). REM soil exhibits a similar fine powder feature with that of raw soil, while the calcined soil particle is severely aggregated (Supplementary Fig. 42). Laser particle size analysis results also reveal comparable size distributions between raw soil and REM soil, but a significant increase of particle sizes with much lower clay and silt ratio[41,42] after calcination (Fig. 4b and Supplementary Fig. 43). Consequently, the calcined soil shows a drastically decreased surface area compared with raw soil and REM soil (Supplementary Fig. 44). The main crystalline composition is quartz for all tested soil samples (Fig. 4c), and XRF results show that no obvious change for various oxides in the soil after treatment (Fig. 4d).

Second, the soil water infiltration rates were assessed. REM soil shows a slightly higher infiltration rate (~34 cm h$^{-1}$) than raw soil (~28 cm h$^{-1}$, Supplementary Fig. 45). Considering a larger porosity of

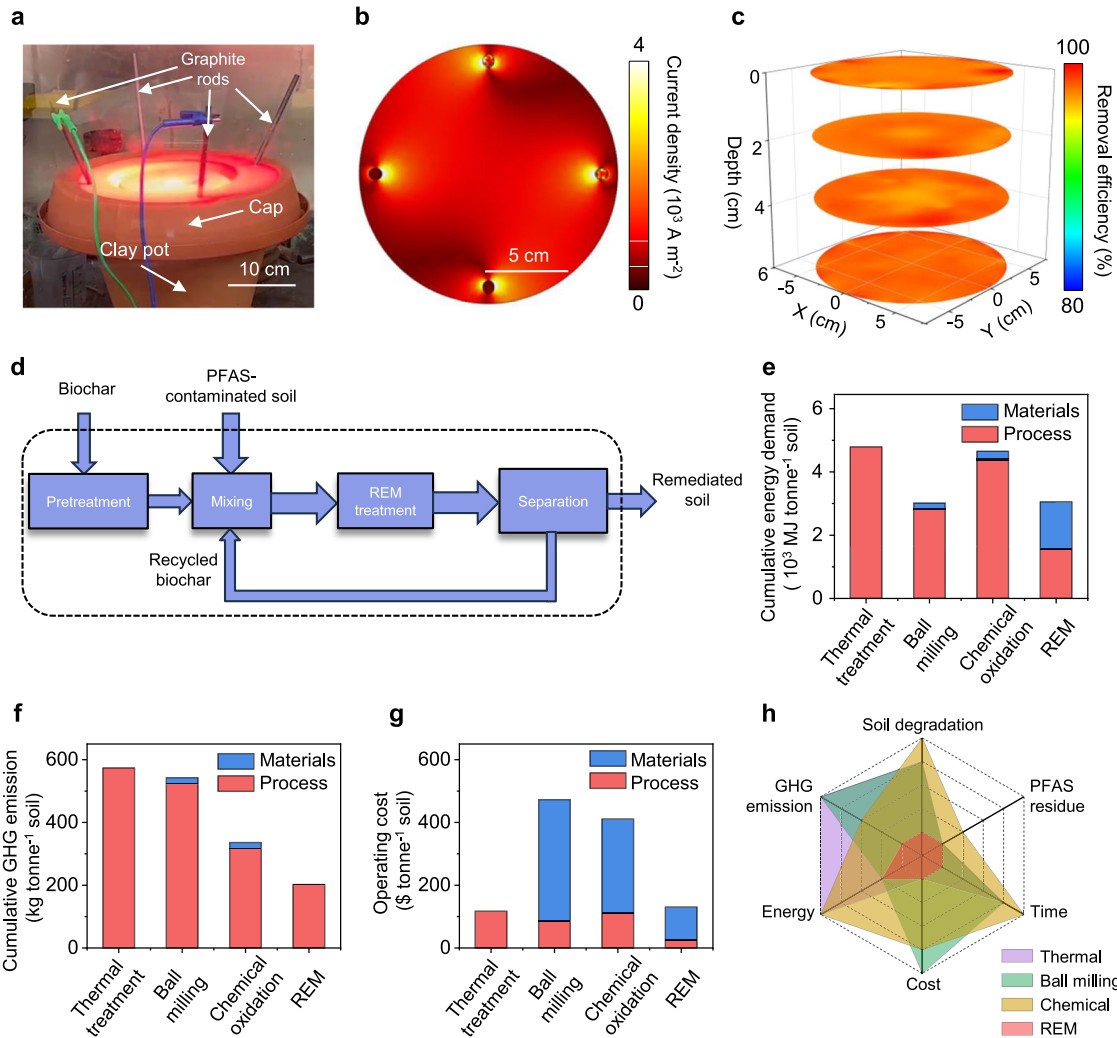

**Fig. 5 | Scalability, LCA, and TEA for the remediation of PFAS-contaminated soil. a** Picture of the kilogram-scale REM process. **b** Simulated distribution of current density on the soil surface with external voltage input. **c** 3D mapping of PFOA removal efficiency. The mapping was sampled from 52 positions of 4 plane depths with an interval of 2 cm and 13 points in each plane. **d** Materials flow analysis of REM. The dash rectangle denotes the system boundary. **e** Comparison of cumulative energy demand. **f** Comparison of cumulative GHG emission. **g** Comparison of operating cost. **h** Comprehensive comparison of different soil-remediation methods.

biochar to soil (Supplementary Figs. 33 and 44), the small amounts of residual biochar in REM soil could contribute to the higher water infiltration rate. In contrast, the infiltration rate of calcined soil (~455 cm h$^{-1}$, Supplementary Fig. 46) is >10 times higher, probably due to its severely enlarged particle size (Fig. 4b and Supplementary Figs. 42 and 43), as water flows faster through the enlarged pores between soil particles[43]. The excessively high infiltration rate would lead to degradation of soil fertility by eluviation[44].

Third, soil pH, cation exchange capacity (CEC), soil carbon, and nutrients contents were analyzed. REM soil exhibits a pH of 7.58, which is slightly higher than that of raw soil (pH = 7.19). The CEC of REM soil is 15.45 cmol kg$^{-1}$, which is comparable to that of raw soil (15.25 cmol kg$^{-1}$, Supplementary Table 6). On the contrary, the pH of calcined soil increases to 10.63 and its CEC decreases to 4.08 cmol kg$^{-1}$ (Supplementary Table 6), indicating the inapplicability for its reuse. Soil carbon content measurement shows that REM soil has a slightly higher carbon content (4.3 wt%) than raw soil (3.7 wt%, Supplementary Fig. 47), possibly due to the existence of a small amount of residual biochar. On the contrary, the calcined soil has a carbon content of <0.1 wt%. The contents of extractable organic compounds, including humic acid and fulvic acid, were quantified by the Walkley–Black

method[45], where <1 wt% of these compounds remained in the REM soil, indicating the decomposition of these compounds during REM process (Supplementary Fig. 48). For the practical use, the organic contents in REM soil can be easily recovered by introducing microorganisms to decompose plant/animal remaining. The exchangeable nutrient content, including P, Ca, K, Mg, Mn, Fe, and N, is a critical parameter to evaluate soil fertility and directly related to soil biodiversity[46,47]. The contents of most exchangeable nutrients in REM soil increased by 10% to 102%, except a ~5% decrease of Fe content (Fig. 4f and Supplementary Figs. 49 and 50). We also evaluated the influence of different carbon additives on soil nutrient contents and found that nutrient-rich biochar can facilitate higher exchangeable nutrient contents of REM soil, comparing with those of the REM soil using metcoke as carbon additive (Supplementary Fig. 51). Therefore, the increase of nutrient contents in REM soil can be attributed to the ion-exchange from biochar (Supplementary Table 7) to the soil, and/or the mineralization of soil organic matter during REM[48,49]. However, most of these nutrient contents dramatically decreased for the calcined soil.

Finally, we conducted the arthropod culture to evaluate the applicability of REM soil in ecosystems. Springtail and isopod were

used as two representative arthropods. We compared the survival ratios of four kinds of soil samples, including PFOA-contaminated soil (denoted as PFOA soil), raw soil, REM soil, and calcined soil (Supplementary Figs. 52 and 53). Because of the toxicity of PFOA[50,51], both springtails and isopods underwent rapid mortality in PFOA soil within the initial 1 to 2 weeks (Fig. 4g, h and Supplementary Tables 8 and 9). In contrast, REM soil exhibited a comparable arthropod survival ratio with raw soil (Fig. 4g, h), demonstrating the effective elimination of toxic substances and the restoration of soil properties. The calcined soil displayed a lessening in arthropods survival ratio compared with raw soil (Fig. 4g, h), which could originate from the nutrient loss and soil structure change. These results reveal that apart from the PFAS mineralization, REM maintains soil morphology, particle size, crystal components and water infiltration rate, while promoting soil nutrients and biodiversity. This is in striking contrast to the calcination process, which leads to soil degradation. This difference can be attributed to the short duration of the REM process that lasts only seconds with rapid heating and cooling rates.

## Scale-up, life cycle assessment, and techno-economic analysis

To outline the practical applicability of REM for PFAS-contaminated soil remediation, we first conducted an initial scale-up study. The PFAS mineralization efficiency depends mainly on the peak temperature and reaction duration during REM. Therefore, maintaining a constant temperature is critical for the scale-up, which can be realized by increasing the input voltage or capacitance of the REM system (Supplementary Note 1). We developed a second-generation REM system with a larger capacitance of $C = 0.624$F (Supplementary Fig. 54). With an input voltage of 300 V, the sample temperature can ramp to 1700 °C, and ~7 g of contaminated soil per batch can be remediated within 6 s. Furthermore, an alternating current (AC) source with better scalability was integrated into a third-generation REM system, which directly converts commercial AC into DC output instead of using capacitors (Supplementary Fig. 55). We mixed 2 kg of PFOA-contaminated soil with 500 g of metcoke in a 10-inch-diameter clay pot with a plastic cap, where four graphite rods were applied as the electrodes (Fig. 5a and Supplementary Fig. 55). During REM, bright light emission was observed through the cap (Fig. 5a), with a steady current of ~18 A and the temperature of ~1000 °C (Supplementary Fig. 55c, d). Afterwards, the soil samples from different positions in the pot were collected for the PFOA quantification (Supplementary Fig. 56). The average PFOA removal efficiency of the kilogram-scale REM reaches ~97% with high uniformity both radically and axially (Fig. 5b), comparable to that of small-scale samples. We further conducted numeric simulation of the current density across the soil under external voltage input (Supplementary Note 2, Supplementary Fig. 57, and Supplementary Table 10). The current density is uniform both in-plane (Fig. 5c) and in-depth (Supplementary Fig. 58), substantiating the homogeneous heating capability of the REM process for soil remediation. The field-scale application potential of REM was evaluated by simulation at a 1 m × 1 m × 1 m scale (Supplementary Fig. 59). Under the voltage input at 1000 V, the current density of the 1 m³ scale soil sample is similar to that in the small-scale clay pot (Fig. 5b). Since the current density mainly determines the accessible temperature (Supplementary Note 1), we project a similar heating pattern can be achieved for the large-scale sample. In addition, the current density of the large-scale soil sample uniformly distributes both in-plane and in-depth (Supplementary Fig. 59d, e), confirming the homogenous heating capability of REM for on-site soil remediation. Meanwhile, based on the simulation results, the increase of electrode surface areas facilitates an increase of current density with a certain voltage input, leading to a higher REM temperature for PFAS mineralization (Supplementary Figs. 60 and 61). For the practical application, considering the moisture contained in field soil, we assessed the applicability of REM for PFAS mineralization in the wet soil. After REM, the wet soil

with a moisture content of ~30 wt% achieved a PFOA mineralization ratio comparable to pre-dried soil (Supplementary Fig. 62), further suggesting the feasibility of REM for practical deployment.

We then assessed the environmental impact of the REM process. The energy consumption of the REM process is calculated to be ~420 kWh ton⁻¹ (Supplementary Note 3). This low energy consumption benefits from the short duration, ultrafast heating/cooling rates, and in-place soil treatment. Furthermore, a comparative cradle-to-gate life cycle assessment (LCA) was conducted to compare the environmental impact and cumulative energy demand of REM with existing remediation approaches (Supplementary Note 4 and Supplementary Tables 11–16). Four scenarios were considered in this study (Fig. 5d and Supplementary Fig. 63), including thermal treatment, chemical oxidation, ball milling, and REM. REM demonstrates a low cumulative energy demand (CED) of 3053 MJ tonne⁻¹. This value is comparable with that of the chemical oxidation process (3013 MJ tonne⁻¹), but 31–33% lower than traditional thermal treatment and ball milling methods (Fig. 5e). REM also exhibits 40–65% reduction in greenhouse gas emission (GHG, Fig. 5f), and 47–67% reduction in water consumption (Supplementary Fig. 64a) compared to other methods. The REM also has no chemical waste generation because of no consumption of chemicals (Supplementary Fig. 64b). Additionally, REM can realize >99% PFAS removal within seconds, achieving the best performance in overcoming the trade-off between removal efficiency and processing time among reported methods[17,18,52–54] (Supplementary Fig. 65).

Finally, a techno-economic analysis (TEA) is conducted, since economic incentives play a vital role in utilization. It is shown that REM has an operating expense of $130 tonne⁻¹ of soil treated, which is comparable to thermal treatment ($117 tonne⁻¹), but much lower than ball milling ($411 tonne⁻¹) and chemical oxidation ($473 tonne⁻¹) (Fig. 5g, Supplementary Note 4 and Supplementary Table 17). With the merits of low cost, high PFAS removal and degradation efficiency, rapid treating process, zero water use, and the preservation of soil properties (Fig. 5h), the REM process shows potential superiorities over existing thermal treatment and chemical oxidation methods toward practical applications.

## Methods
### Materials
The biochar (Wakefield Biochar) was purchased from Amazon. Before mixing with the soil, ~300 mg of biochar in a batch was pretreated by rapid electrothermal process for 1 s with input voltage of 60 V to fulfill the required conductivity as the conductive additive. The equipment for this process was shown in Supplementary Fig. 1. The average size of the pretreated biochar was ~150 μm, and its morphology and size distribution are shown in Supplementary Fig. 33. Carbon black (Cabot, Black Pearls 2000, average diameter ~10 nm) or metallurgical coke (metcoke, SunCoke Energy) were also used as the conductive additives. The used PFAS include perfluorooctanoic acid (PFOA, 95%, Millipore-Sigma), heptadecafluorooctanesulfonic acid tetraethylammonium salt (PFOS, 98%, Millipore-Sigma), tridecafluorohexane-1-sulfonic acid potassium salt (PFH$_x$S, 98%, Millipore-Sigma), nonafluorobutane-1-sulfonic acid potassium salt (PFBS, 98%, Millipore-Sigma), and polytetrafluoroethylene (PTFE, AF2400, Runaway Bike). Raw clean soil was obtained from the Rice University campus. The as-collected soil was crushed by a hammer grinder (Wenling LINDA machinery Corporation, DF-15) and then dried in an oven for 2 h at 100 °C to remove the residual moisture.

### PFAS mineralization by REM process
~1.0 mg PFAS were dissolved in 10.00 g of ultrapure water (Millipore-Sigma, ACS reagent for ultratrace analysis). Then, 10.00 g of raw soil was added, followed by shaking on a horizontal shaker (Burrell Scientific Wrist Action, Model 75) for 24 h and then dried overnight.

The specific mixed PFAS concentration was tested by LC-MS and listed in Supplementary Table 1.

The electrical diagram and picture of the REM system are presented in Supplementary Fig. 1. During the REM process, a mixture of PFAS-contaminated soil (~200 mg) and carbon conductive additives (~100 mg) with a total mass of ~300 mg was loaded into a quartz tube with inner diameter (ID) of 8 mm and outer diameter (OD) of 12 mm after hand-milling for 2 min. Two graphite rods were fixed on each side of the quartz tube as the separators to avoid contamination from the metal electrodes. The tube was loaded on a customized reaction jig and connected to the external REM power system. The two brass electrodes with O-rings were applied to compress and seal the sample inside the tube to prevent the gas emission (Supplementary Fig. 1c). A spring coiled on the surface of the tube was used to avoid the accumulated pressure-induced breaking of the quartz tube during the REM process. The jig was put into a vacuum desiccator under the vacuum of ~10 mm Hg and then connected to the REM system. The capacitor bank (60 mF) was charged by an AC supply and output a DC pulse. The maximal voltage of the capacitor bank can reach 400 V. The relay with programmable delay time with millisecond controllability was applied to control the discharging time. The input voltage was modulated from 0 to 150 V and the discharging time was regularly set as 1 s. The REM temperature was recorded using two IR thermometers (Micro-Epsilon), the detecting range of which are 200–1500 °C and 1000–3000 °C, respectively. These thermometers are connected to LabView using a Multifunction I/O (NI USB-6009) for real-time temperature recording with time resolution of 0.1 ms. After REM, the samples rapidly cooled to room temperature and were collected for further analysis.

To investigate the cation influence on the PFAS mineralization, different alkaline and alkaline earth carbonates, including $CaCO_3$, $MgCO_3$, and $Na_2CO_3$, were mixed with PFAS. The metal counterion content is 1.2 mole equivalent compared with the F mole content in PFAS to ensure complete PFAS mineralization. Metcoke was used as the carbon additives and the total sample mass per batch was set as 300 mg. During REM, the input voltage was set as 100 V, and the discharging time was set as 1 s (See details in Supplementary Table 4).

For the enlarged sample, a mixture of soil (~7 g) and biochar (~4 g) was loaded into a quartz tube with ID of 16 mm and OD of 20 mm (Supplementary Fig. 54b). A second-generation REM system with a larger capacitance of $C = 0.624$ F was applied for REM energy input (Supplementary Fig. 54a).

For the scale-up, 2 kg of soil mixed with 500 g of metcoke inside a clay pot with the outer diameter of 25.4 cm and the soil depth of ~7 cm. Four graphite rods (30 cm in length and 4 mm in diameter) were inserted into the soil as the electrodes. The resistance of each two adjacent electrodes is ~50 Ω. Each two adjacent electrodes were connected to the external power system (Supplementary Fig. 55) sequentially for the uniform treatment of the soil, with 10 s duration time and 4 times REM with each two adjacent electrodes were conducted in total.

### PFAS measurement by HPLC and QQQ LC-MS
200 mg of PFOA-contaminated soil and REM soil were separately added into 2 mL of ultrapure water (Millipore-Sigma, ACS reagent for ultratrace analysis), with the soil-to-water mass ratio of 1:10. The mixture was shaken on a horizontal shaker (Burrell Scientific Wrist Action, Model 75) for 2 h for complete extraction. Then, the sample was centrifuged (Adams Analytical Centrifuge) with the speed of 580 g for 2 min, followed by filtration using polyethersulfone (PES) membrane (0.22 μm, Millipore-Sigma). The PES filter had a negligible influence on PFAS detection (Supplementary Fig. 66). Afterwards, the concentration of PFOA in the extractant was determined by HPLC-DAD (1260 Infinity II Agilent) and a WPH C18 column (4.6 mm × 250 mm, 5 μm),

where the mobile phase was 50% acetonitrile and 50% of 5 mmol $L^{-1}$ $Na_2HPO_4$ with a flow rate of 0.8 mL $min^{-1}$ and 50 μL injection volume for each sample[10]. The calibration curve was prepared by dissolving a known amount of PFOA in ultrapure water (Millipore-Sigma, ACS reagent for ultratrace analysis) in the range from 1 to 100 ppm (Supplementary Fig. 5).

In order to detect the trace amount of residual PFOA after REM (<1 ppm), a QQQ-LC/MS system (6740B LC/TQ, Agilent) using dynamic multiple reaction monitoring (DMRM) was applied. Here, the chromatographic separation was performed on a C18 analytical column (Zorbax Eclipse Pluse C18 Rapid Resolution HT, 2.1 × 50 mm 1.8-μ column, Agilent) with an ultra-high-performance LC (UHPLC) system (1290 Infinity II, Agilent). The aqueous phase consisted of 20 mM ammonium acetate solution, and the organic phase of methanol. The column was operated at a temperature of 40 °C and 40 μL sample was injected each time. The mobile phase flow rate was maintained at 0.4 mL $min^{-1}$ throughout the run. The column was equilibrated at initial conditions for 3 min before the next injection. The LC-MS system was interfaced to the MS system through an Agilent Jet Spray (AJS) electrospray ionization (ESI) source that was operated in the negative ionization mode. The sample preparation procedure is the same with the HPLC-DAD test, and the extractant needs to be further diluted to keep PFOA concentration in the range of 0.5 to 100 ppb. The calibration curve was prepared by dissolving a known amount of PFOA in ultrapure water (Millipore-Sigma, ACS reagent for ultratrace analysis) in the range of 0.5 to 100 ppb (Supplementary Fig. 6).

For detecting other PFAS (PFOS, $PFH_xS$, and PFBS) in the soil by LC-MS, the PFAS extractants were diluted using 90 vol% ultrapure water and 10 vol% methanol (Millipore-Sigma, HPLC standard, >99.9%) to a concentration within the detecting range of 0.5–100 ppb.

The removal efficiency (E) of PFAS was calculated according to Eq. (1),

$$E = \frac{c(\text{Residue PFAS}) \times D_1}{c(\text{Original PFAS}) \times D_2} \times 100\% \qquad (1)$$

where c(Original PFAS) and c(Residue PFAS) are the concentration of PFAS measured by LC-MS before and after REM, $D_1$ and $D_2$ are the dilution factors of PFAS in the raw soil and REM soil, respectively.

### Total fluorine content test for CIC
The soil sample (~10 mg) was loaded into a combustion furnace (AQF-2100H, NITTOSEIKO ANALYTECH) with the temperature of 1100 °C under 400 mL $min^{-1}$ oxygen flow. The combusted anions were absorbed by 100 mL $min^{-1}$ water-saturable Ar and 200 mL $min^{-1}$ Ar, and then flowed into a gas absorption unit (GA 211, Mitsubishi Chemical Analytech). Afterwards, total F content was analyzed by an IC system (Dionex ICS-2100, Thermo Scientific).

### Inorganic fluoride measurement by IC
200 mg of raw soil and REM soil were separately added into 4 mL of ultrapure water (Millipore-Sigma, ACS reagent for ultratrace analysis), with the soil to water mass ratio of 1:20. Then, the mixture was immersed into an ultrasonic bath (Cole-Parmer Ultrasonic Cleaner) for 15 min for the extraction, followed by centrifugation (Adams Analytical Centrifuge) with the speed of 580 g for 2 min, and filtration using PES membrane (0.22 μm, Millipore-Sigma) to remove any undissolved particles. The concentration of mineralized fluorine ion in the sample was measured by IC (Dionex Aquion, 4 × 250 mm IonPac AS23, AERS 500 Carbonate Suppressor). The calibration curve was prepared by dissolving a known amount of sodium fluoride in ultrapure water (Millipore-Sigma, ACS reagent for ultratrace analysis) in the range of 1 to 5 ppm (Supplementary Fig. 7).

The mineralization ratio ($R$) of PFAS is calculated according to Eq. (2),

$$R = \frac{c(F - ion) \times r \times D_1}{c(PFAS) \times D_2} \times 100\% \qquad (2)$$

where $c$(PFAS) is the concentration of PFAS measured by LC-MS, $c$(F-ion) is the concentration of fluorine ions measured by IC, $r$ is the mass ratio of fluorine atom in a certain PFAS molecular (listed in Supplementary Table 1), and $D_1$ and $D_2$ are the dilution factors of PFAS and fluorine ions, respectively. Note for the PTFE, $c$(PTFE) was calculated by dividing initial F content from CIC data by $r$(PTFE)

## NMR test
1 g of PFAS-contaminated soil and REM soil were separately added into 2 mL of deuterium oxide ($D_2O$, 99.9%, Millipore-Sigma). The mixture was immersed into an ultrasonic bath (Cole-Parmer Ultrasonic Cleaner) for 15 min for the extraction, followed by centrifugation (Adams Analytical Centrifuge) with the speed of 580 g for 2 min, and filtration using a PES membrane (0.22 μm, Millipore-Sigma) to remove any undissolved particles. The solution was then added into the NMR tube and the chemical shift was tested by NMR spectrometer (600 MHz Bruker NEO Digital NMR Spectrometer).

## GC-MS test
To study the evolved gases, REM was carried out in a home-designed jig. The evolved gas can vent from the quartz reaction tube through a hollow electrode into a sealed tube with pressure gauge (Supplementary Fig. 11). The REM parameters were kept same as mentioned above. The system was purged with argon gas, and evacuated to −75 kPa before REM. After REM, the evolved gases were injected into the GC-MS using a gas-tight syringe. The GC-MS instrument used here was an Agilent 8890 GC system equipped with an Agilent HP-5ms low-bleed column (30 m, 0.25 mm internal diameter, 0.25 μm film) with helium as the carrier gas for liquid and headspace sampling. A tandem Agilent 5977B mass selective detector was used for liquid and headspace gas analysis. For the gas detection, the injector and the transfer line were set with the temperature of 120 and 200 °C, respectively. The temperature program was initiated at 48 °C for 3 min, and then increased to 80 °C at 8 °C min⁻¹. The carrier gas was helium at a flow rate of 0.5 mL min⁻¹. For the F-containing residue detection, ~200 mg REM treated soil samples were added into 5 mL extractant solvent (mixture of hexane, acetone, and toluene with volume ratio of 10:5:1). Then, the mixture was immersed into an ultrasonic bath (Cole-Parmer Ultrasonic Cleaner) for 15 min for the extraction, followed by centrifugation (Adams Analytical Centrifuge) with the speed of 580 g for 2 min, and filtration using PES membrane (0.22 μm, Millipore-Sigma) to remove any undissolved particles. The filtered solution was loaded onto a GC autosampler. During the test, the injector and the transfer line temperature were set to 280 and 300 °C, respectively. The temperature program was initiated with 75 °C for 1 min, increased to 230 °C at 10 °C min⁻¹ held for 7 min, then to 280 °C at 20 °C min⁻¹, and held for 15 min. The injection volume was 1 μL each time in a splitless mode, and solvent delay was 5 min to prevent filament damage. The carrier gas was helium at a flow rate of 1.2 mL min⁻¹.

## Biochar recycling by centrifugation
In all, 500 mg of REM soil mixed with biochar was dispersed into 5 mL of water (Millipore-Sigma, ACS reagent for ultratrace analysis), followed by shaking on horizontal shaker (Burrell Scientific Wrist Action, Model 75) for 15 min. The mixture was then centrifugated (Adams Analytical Centrifuge) with a speed of 580 g for 2 min. After centrifugation, the lightweight biochar floated on the water, while the dense soil particles sank (Supplementary Fig. 19). The floating biochar was then poured and filtered using a sand core funnel (Class F). The

separated soil and biochar were then separately dried in an oven at 100 °C for 2 h to remove the residual moisture.

## Soil calcination
In total, ~10 g of PFOA-contaminated soil placed in an alumina crucible was heated in a muffle furnace (Carbolite RHF 1500). The sample temperature increases to 900 °C with the heating rate of 20 °C min⁻¹ and maintained at 900 °C for 2 h in the air. Afterwards, the sample naturally cools to room temperature.

## Infiltration rate test
A quartz tube with an ID of 16 mm was used as the container with a sponge to hold the soil samples, enabling fast water penetration. Water can penetrate the sponge rapidly thus does not affect the infiltration rate. The raw soil, REM soil, and calcined soil with the same volume were separately placed on top of the sponges, and 2 cm of water was then gently added atop the soil. The liquid level was measured at different times, and the water infiltration rate was calculated by Eq. (3),

$$\text{infiltration rate} = H/t \qquad (3)$$

where $H$ is the liquid level in cm and $t$ is the time in min (Supplementary Figs. 45 and 46).

## Particle size measurement
To prepare the samples, we separately added 1.0 g of raw soil, REM soil, calcined soil, or biochar into 10.0 mL of 0.1 M HCl solutions. The carbonate inside the soil was removed by reacting with HCl under an ultrasonic bath (Cole-Parmer Ultrasonic Cleaner) for 15 min. Then, the samples were centrifuged (Adams Analytical Centrifuge, 580 g, 5 min) and washed three times with ultrapure water (Millipore Sigma, ACS reagent for ultratrace analysis). Next, 2.0 mL of $H_2O_2$ solution (Millipore Sigma, ~35 wt%) was mixed with the soil in a 90 °C water bath for 45 min to remove the soil organic matter[55]. After another round of centrifugation and water washing three times, 1 g of soil particles were dispersed in 5 mL of water solution with 3.3 wt% sodium hexametaphosphate and 0.7 wt% sodium carbonate. Afterwards, it was injected into a laser particle size analyzer (ZEN 3600 Zetasizer Nano, Malvern, Worcestershire, UK) for particle size measurement. According to the measured data, we further counted the ratio of clay (<2 μm), silt (2–50 μm), and sand (>50 μm) in the soil based on the particle size distribution.

## Soil carbon content measurement
The soil carbon content was measured using an ECS 4010 – CHNS-O Elemental Combustion System. Before the measurement, 1.0 g of soil sample was dispersed into 10.0 mL of 0.1 M HCl in an ultrasonic bath (Cole-Parmer Ultrasonic Cleaner) for 15 min to remove carbonate. Then, the samples were washed three times with ultrapure water (Millipore Sigma, ACS reagent for ultratrace analysis). Afterwards, the sample dried at 105 °C. Acetanilide was used as the standard material for calibration. Raw soil, REM soil, and calcined soil were subjected to carbon content measurement. Each sample was tested in triplicate to obtain the standard deviations.

## Exchangeable nutrients measurement
The exchangeable P, Ca, K, Mg, Mn, and Fe in raw soil, REM soil, and calcined soil were extracted using the Mehlich-3 reagent[56]. The extractant is composed of 0.2 M $CH_3COOH$, 0.25 M $NH_4NO_3$, 0.015 M $NH_4F$, 0.013 M $HNO_3$, and 0.001 M ethylenediaminetetraacetic acid (EDTA). 1 g of soil sample was added to 10 g of the extract with a soil to solution ratio mass of 1:10. The mixture was shaken immediately on a horizontal shaker (Burrell Scientific Wrist Action, Model 75) for 15 min. Then, the sample was centrifuged (Adams Analytical Centrifuge, 580 g, 5 min), followed by filtration using a PES membrane (0.22 μm,

Millipore-Sigma) to remove any undissolved particles. The filtrate was diluted to appropriate concentration using 2 wt% $HNO_3$ within the calibration curve range. The P, K, Mg, Mn, and Fe were measured by inductively coupled plasma mass spectrometry (ICP-MS) using a Perkin Elmer Nexion 300 ICP-MS system, with Periodic Table mix 1 for ICP (10 mg L$^{-1}$, 10 wt% $HNO_3$, Millipore Sigma) as the standard. Due to interference from Ar, Ca cannot be measured by ICP-MS. Therefore, Ca was measured by inductively coupled plasma optical emission spectrometer (ICP-OES) using a Perkin Elmer Optima 8300 ICP-OES system. Ca standard (1000 mg L$^{-1}$, 2 wt% $HNO_3$, Millipore Sigma) was used for the ICP-OES measurement.

The soil nitrate-nitrogen serves as an indicator of available nitrogen for plants. The soil nitrate content was measured using IC (Dionex Aquion, 4 × 250 mm IonPac AS23, AERS 500 Carbonate Suppressor). Nitrate standard calibration solutions were prepared by dissolving $NaNO_3$ in ultrapure water (Millipore Sigma, ACS reagent for ultratrace analysis) in the concentration range from 0.5 to 15 ppm. The good linearity of the calibration curve demonstrates the effectiveness of this method (Supplementary Fig. 49). To extract nitrate in raw soil, REM soil and calcined soil, 1 g of soil sample was separately added into 10 g of ultrapure water (Millipore Sigma, ACS reagent for ultratrace analysis) for the nitrate extraction and sonicated in an ultrasonic bath (Cole-Parmer Ultrasonic Cleaner) for 15 min. Then, the sample was centrifuged (Adams analytical centrifuge, 580 g, 5 min), followed by filtration using PES membrane (0.22 μm, Millipore-Sigma) to remove any undissolved particles. Finally, the filtrate was diluted to a concentration within the calibration curve range of 0.5 to 15 ppm.

### Soil CEC test

1.0 g of soil sample was dispersed into 10 mL of 1 M sodium acetate solution in an ultrasonic bath (Cole-Parmer Ultrasonic Cleaner) for 15 min to saturate soil exchange sites with $Na^+$. Then, the sample was washed three times with ethanol (Decon's Pure 200 Proof, Decon Labs Inc.) to remove the excess $Na^+$. Afterwards, the sample was dispersed into 10 mL of 1 M ammonium acetate in an ultrasonic bath for another 15 min to replace $Na^+$ by $NH_4^+$ at exchange sites[55]. The sample was then centrifuged (Adams analytical centrifuge, 580 g, 5 min), followed by filtration using PES membrane (0.22 μm, Millipore-Sigma) to remove any undissolved particles. The filtrate was diluted to the appropriate concentration using 2 wt% $HNO_3$ within the ICP calibration curve range. The $Na^+$ concentration was measured by the Perkin Elmer Nexion 300 ICP-MS system, with sodium standard solution for ICP (1 g L$^{-1}$, Millipore Sigma) as the standard. Finally, CEC was calculated by Eq. (4),

$$CEC = \frac{c(Na^+) \times D \times V}{M(Na^+) \times m(soil)} \quad (4)$$

where $c(Na^+)$ is the concentration of sodium measured by ICP-MS, $D$ is the dilution factor, $V$ is the volume of ammonium acetate solution to extract $Na^+$ ($V = 10$ mL), $M(Na)$ is mole mass of sodium ($M(Na^+) = 23$ g mol$^{-1}$), $m(soil)$ is the mass of soil sample used for CEC test ($m(soil) = 1.0$ g).

### Soil organic content test

In all, 1.0 g of soil sample was dispersed into 10 mL of extractant ($V_1 = 10$ mL), composed of 0.5 M NaOH and 0.5 M $Na_4P_2O_7$. The mixture was shaken on a horizontal shaker (Burrell Scientific Wrist Action, Model 75) for 1 h, and then heated at 95 °C for 30 min, followed by centrifuging (Adams Analytical Centrifuge, 580 g) for 2 min. After filtrating through a PES membrane (0.22 μm, Millipore-Sigma), we obtained solution 1. For the total organic content test, 1 mL of solution 1 ($V_2 = 1$ mL) was mixed with 5 mL of 0.4 M $K_2Cr_2O_7$ and 15 mL of 2 M $H_2SO_4$, and then heated at 95 °C for 30 min to oxidize the organic compounds in the extractant. After cooling to room temperature, the

solution was mixed with 78.9 mL ultrapure water (Millipore Sigma, ACS reagent for ultratrace analysis) and 0.1 mL of phenanthroline indicator (1.5 wt% phenanthroline and 1 wt% $(NH_4)_2Fe(SO_4)_2$), which was denoted as solution 2. Afterwards, 0.1 M $(NH_4)_2Fe(SO_4)_2$ was gradually added to solution 2 until the solution's color changed from orange to green and finally to brick red. The consumed volume of $(NH_4)_2Fe(SO_4)_2$ solution was recorded as $V_3$. For the comparison, 0.1 M $(NH_4)_2Fe(SO_4)_2$ was gradually added to a solution with 5 mL of 0.1 M $K_2Cr_2O_7$, 15 mL of 2 M $H_2SO_4$, 74.9 mL of ultrapure water (Millipore Sigma, ACS reagent for ultratrace analysis) and 0.1 mL of phenanthroline indicator. The consumed volume of $(NH_4)_2Fe(SO_4)_2$ solution was recorded as $V_0$ when the solution color changed to green. Therefore, the total organic mass content ($c_{org}$, with the unit of g kg$^{-1}$) can be calculated from Eq. (5):

$$c_{org} = \frac{0.003 \times (V_0 - V_3) \times c(Fe^{2+}) \times r_o \times r_c}{m} \times \frac{V_1}{V_2} \times 1000 \quad (5)$$

Where $c(Fe^{2+})$ is the mole concentration of $(NH_4)_2Fe(SO_4)_2$ ($c(Fe^{2+}) = 0.1$ M), $m$ is the soil mass ($m = 1$ g), $r_o$ and $r_c$ is the oxidation factor and the conversion factor from organic carbon to organic compound ($r_o = 1.1$ and $r_c = 1.724$).

Considering the insolubility of humic acid in acid solution, 2 M $H_2SO_4$ was added to 5 mL of solution 1 ($V_5 = 5$ mL) until the pH reached 1 and it was then left for 30 min to separate humic acid from the soil extractant. After filtering using a sand core funnel (class F), the filter residue was washed by 0.05 M $H_2SO_4$ for 5 times. Afterwards, the residue was dissolved by 50 mL 1 wt% NaOH and then diluted to 100 mL using ultrapure water (Millipore Sigma, ACS reagent for ultratrace analysis), which is denoted as Solution 3 ($V_6 = 100$ mL). Similarly, 5 mL of 0.4 M $K_2Cr_2O_7$ and 15 mL of 2 M $H_2SO_4$ were used to oxidize 5 mL Solution 3 ($V_7 = 5$ mL) at 95 °C for 30 min and the residual $K_2Cr_2O_7$ in solution was titrated by 0.1 M $(NH_4)_2Fe(SO_4)_2$ using the phenanthroline indicator. The consumed volume of $(NH_4)_2Fe(SO_4)_2$ solution was recorded as $V_8$. The humic acid mass content ($c_{humic}$, with the unit of g kg$^{-1}$) can be calculated from Eq. (6):

$$c_{humic} = \frac{0.003 \times (V_0 - V_8) \times c(Fe^{2+}) \times r_o \times r_c}{m} \times \frac{V_1}{V_5} \times \frac{V_6}{V_7} \times 1000 \quad (6)$$

The fulvic acid content ($c_{fulvic}$, with the unit of g kg$^{-1}$) can be thus calculated by:

$$c_{fulvic} = c_{org} - c_{humic} \quad (7)$$

### Arthropod culture

For the isopod culture, we performed lab microcosm experiments where 2 adult *Armadillidium vulgare*, a common isopod species, was added to Petri dishes (35 mm diameter, 10 mm height). Four different kinds of soil treatments were tested for isopods, including (1) raw soil, (2) PFOA-contaminated soil, (3) REM soil, and (4) calcined soil. Isopod culture for each soil sample was replicated 7 times. Before the experiment, 1.5 g of the soil sample was added to the Petri dishes and the isopods were hand-picked from a lab-reared culture. Water and isopod food were added every other day to ensure the high survival of isopods. The isopod microcosm experiment was conducted in a light and humidity incubator at 25 °C and with a 12-h night and day cycle. Isopod survival ratio was measured weekly for 4 weeks.

A similar microcosm experiment was employed to evaluate the springtails survival ratio in different kinds of soil samples. Four different soil samples for springtail culture include (1) raw soil, (2) PFOA-contaminated soil, (3) REM soil, and (4) calcined soil. *Folsomia candida*, cataloged as an Isotomidae family member and a Collembola species, is used as a representative type of springtail for the culture. Springtail culture for each soil sample was replicated 7 times. Before

the experiment, 1.5 g of the soil sample was added to the Petri dishes (35 mm diameter, 10 mm height). Because of the small size of springtails (< 2 mm), it is difficult to place the exact same number of springtails into each Petri dish. Thus, we used an aspirator to collect springtails from a lab culture and added 10-15 springtail adults to each dish to inoculate the Petri dishes with springtails. To create suitable conditions for springtails, 1.5 mL of deionized water and ~10 mg of food (i.e., Baker's Yeast) were replenished every week to each dish. Springtails survival ratio was measured weekly for 4 weeks.

The number of remaining isopods and springtails were counted every week throughout the experiment. This process was continued for 28 days (i.e., 4 total weeks with 3 days of counting). We used the survival ratio after each week as the response variable for both isopods and springtails. To specifically test how the soil treatments influence arthropod survival, we employed generalized linear models (GLMs, ref. 57) with two fixed factors: (1) soil types and (2) culture time. We used GLMs with a logit binomial distribution and a Poisson distribution for isopods and springtails, respectively, because of higher data consistency. GLMs were fitted using the "lme4" package in R version 4.3.1 software.

## Theoretical calculation

In the MD simulation, PFOA molecules were mixed with CaO in a supercell, where the periodic boundary condition was applied in all three dimensions. Four different atomic ratios of F over Ca were set by changing the number of PFOA and CaO molecules. DFT method[58] was used as implemented in the Vienna Ab-initio Simulation Package (VASP)[59] with climbing image nudged elastic band method (CiNEB)[60]. A plane wave expansion up to 500 eV was employed in combination with an all-electron-like projector augmented wave (PAW) potential[61]. Exchange-correlation was treated within the generalized gradient approximation (GGA) using the functional parameterized by Perdew-Burke-Ernserhof [62]. With the smallest one being 15.0 Å × 15.0 Å × 22.0 Å, all supercells are big enough. Thus, only Γ point was used for the Brillouin zone integration over Monkhorst-Pack type mesh[63]. For the structure optimization using the conjugate-gradient algorithm as implemented in VASP, both the positions of atoms and the unit cells were fully relaxed, so that the maximum force on each atom was smaller than 0.01 eV Å$^{-1}$. For modeling of mineralization reaction, the optimized structures were subsequently annealed for 30 ps with the temperature fluctuating at the range of 1500–2500 K in MD simulation. The MD simulation was performed using Nose-Hoover thermostat and NVT ensemble with a time step of 1 fs. Then, we use the number of F-C bonds in the system as a descriptor of the mineralization effect. The number of unbroken F-C bonds were calculated every 20 steps in each of the MD simulation and the results are shown in Fig. 4f and Supplementary Fig. 39. For counting the number of F-C bonds, the cut-off distance was set at 1.55 Å, as compared to the equilibrium F-C bond-length of 1.44 Å.

The calculations of the Gibbs free energy change under different temperatures were conducted using the HSC Chemistry 10 software.

## Other characterizations

SEM images and element analysis by EDS were taken on the FEI Quanta 400 ESEM FEG system under the voltage of 10 kV and the working distance of 10 mm. XRD was performed using the Rigaku SmartLab system with a filtered Cu Kα radiation ($\lambda = 1.5406$ Å). The FT-IR spectra were acquired on the Thermo Scientific Nicolet 6700 attenuated total reflectance Fourier transform infrared (ATR-FTIR) spectrometer (Waltham, MA, USA). Raman spectra were obtained on the Renishaw Raman microscope system (laser wavelength: 532 nm, laser power: 5 mW, lens: 50×). XPS spectra were conducted using the PHI Quantera XPS system under the pressure of $5 \times 10^{-9}$ Torr. The survey spectra were collected with the step of 0.5 eV and the pass energy of 140 eV, and elemental spectra were collected with the step size of 0.1 eV and

the pass energy of 26 eV. All XPS spectra were calibrated using the C 1s peak at 284.8 eV as the reference. TGA was conducted on the Mettler Toledo TGA/DSC 3+ system using a 70 μL pan with the heating rate of 10 °C min$^{-1}$ and under 100 mL min$^{-1}$ air flow. The TGA system was connected to a mass spectrometer (PrismaPro Quadrupole, Pfeiffer Vacuum) for the TGA-MS test with 100 mL min$^{-1}$ nitrogen as the carrier gas and the heating rate of 10 °C min$^{-1}$. BET measurements were performed on a Quantachrome Autosorb-iQ3-MP/Kr BET surface analyzer at 77 K, where the nitrogen was used as the adsorption/desorption gas.

XRF spectra were acquired by a Panalytical Epsilon 4 XRF instrument. Before test, the soil samples were fused into glass beads using lithium metaborate/lithium tetraborate and lithium nitrate as the fluxing agents using a Katanax K2 Prime instrument. Samples were heated in platinum crucibles to 1000 °C for 15 min while being rocked back and forth for dispersion. After fusion, the platinum crucibles containing the samples were poured into the platinum mold to form beads. The fused beads were then automatically fed into the XRF via the sample loader for continued analysis. The SuperQ analytical software used the documented weights of each sample and its flux weight to generate molar quantitative results. Soil pH was measured by a soil pH meter (SOILPHU), where the detector was inserted into 10 g of soil. The average pH values and the standard deviations for each sample were calculated after 5 times of testing.

## Data availability

The data supporting the findings of the study are included in the main text and supplementary information files. The source data generated in this study have been deposited in the Zenodo database under accession code DOI link. [https://doi.org/10.5281/zenodo.11372476]. Source data are provided with this paper.

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

## Acknowledgements

The authors thank Dr Bo Chen from Rice University for the helpful discussion on XPS results, Dr Christopher Pennington from Rice University for developing ICP and QQQ LC-MS methods, and Dr Helge Gonnermann from Rice University for allowing us to use the FEM simulation software. The funding of the research is provided by Rice Academy Fellowship (Y.C.), Air Force Office of Scientific Research (FA9550-22-1-0526, J.M.T.) and the U.S. Army Corps of Engineers, ERDC grant (W912HZ-21-2-0050, J.M.T., B.I.Y., M.G.U.A., and C.G.). Computer resources were provided through DOE's NERSC award BES-ERCAP0027822. Permission to publish was granted by the Director, Geotechnical & Structures Laboratory, ERDC. The characterization equipment used in this project is partly from the Shared Equipment Authority (SEA) at Rice University.

## Author contributions

Y.C., B.D., and J.M.T. conceived the idea and designed the experiments. Y.C. conducted the sample preparation, REM treatment, and most of the characterizations with the help of J.C., Q.L., and S.X.; Y.C. conducted the IC, HPLC, and LC-MS measurements with the assistance of P.S., B.W., and M.S.W.; B.L. and T.S. assisted with the NMR tests. K.J.S. and K.M.W. assisted with the GC-MS tests. M.G.U. and C.G. conducted the XRF test. Y.C. and L.E. conducted the scale-up experiments with the help of K.J.; A.H. and M.A.M. conducted the arthropods culture. Y.C. and B.D. measured the soil carbon content with the help of X.G.; Y.C. measured the particle size with the help of D.J. and M.A.T.; Y.Z. and B.I.Y. conducted the DFT simulation. Y.C. conducted the thermodynamic analysis and analyzed the DFT simulation results with the assistance of Y.Z.; B.D. conducted the electric field simulation. Y.C. conducted the LCA and TEA. Y.C., B.D., Y.Z., and J.M.T. wrote and edited the manuscript. All aspects of the research were overseen by J.M.T. All authors discussed the results and commented on the manuscript.

## Competing interests

Rice University owns intellectual property on the rapid electrothermal mineralization strategy for PFAS mineralization and soil remediation. A US provisional patent was filed by Rice University, where Y.C., B.D., and J.M.T. are listed as the inventors (United States Provisional Application No. 63/589,489), which has not yet been licensed. The authors declare no other competing interests.

## Additional information

[1]Department of Chemistry, Rice University, Houston, TX, USA. [2]Department of Materials Science and NanoEngineering, Rice University, Houston, TX, USA. [3]Applied Physics Program, Rice University, Houston, TX, USA. [4]Smalley-Curl Institute, Rice University, Houston, TX, USA. [5]Department of Biosciences, Rice University, Houston, TX, USA. [6]Nanosystems Engineering Research Center for Nanotechnology Enabled Water Treatment (NEWT), Houston, TX, USA. [7]Department of Chemical and Biomolecular Engineering, Rice University, Houston, TX, USA. [8]U.S. Army Engineer Research & Development Center, Vicksburg, MS, USA. [9]Department of Earth, Environmental, & Planetary Sciences, Rice University, Houston, TX, USA. [10]Carbon Hub, Rice University, Houston, TX, USA. [11]Department of Civil and Environmental Engineering, Rice University, Houston, TX, USA. [12]Corban University, Salem, OR, USA. [13]NanoCarbon Center and the Rice Advanced Materials Institute, Rice University, Houston, TX, USA. [14]Present address: School of Environment, Tsinghua University, Beijing, China. [15]These authors contributed equally: Yi Cheng, Bing Deng. ✉e-mail: dengbing@tsinghua.edu.cn; YZhao@corban.edu; tour@rice.edu

