## [Peer Review File · Nature Communications]

Electrothermal mineralization of per- and polyfluoroalkyl substances for soil remediationREVIEWER COMMENTS

Reviewer #1 (Remarks to the Author):

Yi Cheng et al., developed a rapid electrothermal mineralization (REM) in the presence of biochar process to remediate PFAS-contaminated soil. The soil temperature increases to >1000 °C within seconds by direct current pulse input, converting PFAS to calcium fluoride with inherent calcium compounds in soil. The general electrical mineralization process is applicable for remediating various PFAS contaminants in soil, with high removal efficiencies (>99.9%) and mineralization ratios (>90%). While retaining soil particle size, composition and water infiltration rate, REM facilitates an increase of exchangeable nutrient supply and arthropod survival in soil. REM has a significant reduction of energy consumption and greenhouse gas emission over existing soil remediation practices. This study of interest and can be consider for publication after major revision.

Some comments

1. In abstract, there is no mention of the scalability or cost-effectiveness of the REM process compared to conventional methods. Evaluating the scalability, economic feasibility, and practical implementation of this technique on a larger scale is necessary for its real-world applicability.
2. In introduction authors briefly discusses stabilization, chemical oxidation, and thermal treatment methods but lacks a detailed comparative analysis of their advantages, limitations, and efficiencies. Providing a more in-depth discussion and comparison of these methods would provide better context for why REM stands out. Take a look at this reference *Current Opinion in Chemical Engineering* 2023, 42:100954
3. While the manuscript briefly touches on the environmental impact, a more comprehensive discussion or comparison of the environmental impact of REM versus other methods would be beneficial. This could include not only energy consumption but also waste generation and the potential long-term effects on soil health and ecosystems.
4. The manuscript mentions the successful remediation on a kilogram scale per batch but lacks discussion on the scalability and practicality of implementing REM on a larger, field-scale level. Further insights into the challenges, cost-effectiveness, and scalability for large-scale deployment would strengthen the manuscript.
5. In Figures 1d and e, at 100 volts, the temperature during the first second of rapid thermal processing could potentially rise to 1300°C or even higher. However, the author discovered that the most effective mineralization occurred at 1000°C. Despite the expectation that higher temperatures might yield better mineralization, the author shall provide an explanation to the reader as to why this optimum temperature was chosen over even higher temperatures for enhanced mineralization.
6. Metcoke seem to appear as a more effective conductive additive when compared to biochar, demonstrating its advantage as approximately 91% by weight can be easily recycled after Rapid Thermal Processing (REM) through a simple sieving method. How author justify the use of biochar.
7. In lines 176-180, the author should explain on how Ca²⁺ facilitates mineralization. They should delve into its chemistry and explain how it potentially impacts the reduction of C-F bond energy.

8. Would other alkaline metals such as Mg or alkali metals potentially facilitate mineralization more effectively? The author should consider exploring this aspect as well.
9. Has the author identified the formation of any side short-chain degradation products in the absence of Ca?
10. How does the author elaborate on the setup's capacity to effectively mineralize PFAS-contaminated soil—up to 99%—and its correlation with the electrodes' surface area in terms of the amount (in kilograms) processed by the current setup?
11. What would be the optimal ratio of biochar or metacoke in relation to a specified quantity of soil? How it normally investigate
12. Line 289 How author calculate the energy consumption of the REM process is calculated to be ~420 kWh ton⁻¹

Reviewer #2 (Remarks to the Author):

In this work, the authors reported a rapid electrothermal mineralization (REM) process to remediate PFAS-contaminated soil. PFAS are known as hard-to-degrade chemicals. Current methods make it difficult to achieve high removal efficiency and defluorination efficiency simultaneously. In this study, the author utilized the REM process to achieve a good PFAS removal effect and defluorination effect. The superiority of REM over various kinds of PFAS removal methods was presented, and the reaction mechanisms were thoroughly discussed. It furnishes us with new ideas for dealing with environmental PFAS in future work. However, there are a few issues that need to be addressed before I recommend its publication in Nature Communications The specific comments are as follows.

- Q1. In line 122, the authors mentioned: “two O-rings on each electrode to seal the reacting tube during REM”. To confirm the necessity of the sealed system, the authors are supposed to supplement more experiments to show the trends of PFOA mineralization efficiency varied with the input voltage in the open system without an O-ring.
- Q2. In Figure 4, the soil properties between raw soil, REM soil, and calcined soil were compared. However, the residual PFOA content and F mineralization content in calcined soil should be supplemented to avoid interference from residue PFOA.
- Q3. In the Methods section, the soil was dried before REM, but most of the soil in nature contains a certain amount of moisture. Can PFOA in wet soil also be effectively removed by the REM method? In other words, is pre-drying a required step for the contaminated soil before the REM process?
- Q4. In the Methods part, the authors mentioned a filtration process using a PES filter before further LC-MS tests. I wonder if the membrane can absorb PFAS during filtration. More experiments are needed to compare the PFAS content in the supernatant and filtered leaching solution.
- Q5: The authors just listed the temperature profiles in Figure 1d. More temperature-time curves for different voltages, like 80 V, 120 V, and 150 V should be provided.
- Q6. In Supplementary Fig. 31, the authors mentioned “the particles with size <2 μm are cataloged as clay, while the size in the range of 2-50 μm for silt, and the size >50 μm for sand.” More

references here need to be provided for the soil classifications.

Q7. There are some spelling mistakes in the paper. For example, Line 101, "Universtisy" should be corrected to "University". please check the entire text.

Reviewer #3 (Remarks to the Author):

Dear Editor

this is a very interesting communication. The study describes how soil is cleaned up from PFAS by electrothermal heating. With low energy and very fast the fluorine from PFAS can be mineralised into benign calcium fluoride by not changing the soil properties significantly. The authors made an astonishing multi-platform analytical effort to show that not only PFAS was reduced to fluoride but that the soil properties did not change significantly. The method is well described and the different analytical methodologies used have sufficient information to redo the experiments - hence transparent enough.

This is the first time that I have seen results of a clean up method for PFAS which can actually work on a larger scale. Hence, that this study could have a significant effect on environmental science. Hence I would recommend to publish this study in Nature Comm. after a successful rebuttal of those queries.

There are however some queries to be answered. The effectiveness was mainly studied first with PFOA and other compounds, but polymers were not so much featured in the study (although PTFE was shown to be degraded in soil by IR and XRD).

The authors should however also give for each method especially IR, ¹⁹F-NMR and XRD the limit of detection in absolute amount as of in ng/g in order to better interpret the graphs and please give also the numbers in a table. This would help to interpret the analytical results. Are these values appropriate for environmental applications? could we detect relevant concentrations?? and can we reduce the PFAS concentration to low enough concentrations for todays regulations??

I do not have experience in XRD, but I do not understand that all other minerals in the soil (and I guess that they are not all amorphous) can be cancelled out to show only the PFAS?? Here a better explanation should be given.

Maybe I missed it but what I need to see is a CIC analysis given the EOF which is often used to identify all extractable organofluorine in a sample. This would be rather interesting especially for PTFE in soil.

The LC-MS method is not appropriate in all cases here. It should be an LC-MS/MS method or even better an LC-HRMS for non-targeted analysis to identify all PFAS and not only the target methods.

The reviewer would like to see an LC-HRMS before and after a an F-containing polymer has been degraded.

Soil properties: what is about the organic content? Cation exchange capacities (CEC), humic and fluvic fulvic acid. They much also be degraded by this treatment?? What happens here?? Corg, how much organic carbon is actually left and in which form. This is crucial!

Dear Reviewers,

We greatly appreciate your conscientious reviews of our manuscript, which are beneficial for strengthening our work. We hope to answer all your questions here, and point-by-point responses to all the questions are listed below. If we unintentionally left something out, we shall correct that if you point it out. If you have further questions, we are eager to address those as well.

The reviewers' comments were copied in *italics*, our responses are in **bold**, and the revisions made to the manuscript are marked in **blue and bold**.

Sincerely,

James M. Tour,

On behalf of all authors.

Reviewer #1 (Remarks to the Author):

Yi Cheng et al., developed a rapid electrothermal mineralization (REM) in the presence of biochar process to remediate PFAS-contaminated soil. The soil temperature increases to >1000 °C within seconds by direct current pulse input, converting PFAS to calcium fluoride with inherent calcium compounds in soil. The general electrical mineralization process is applicable for remediating various PFAS contaminants in soil, with high removal efficiencies (>99.9%) and mineralization ratios (>90%). While retaining soil particle size, composition and water infiltration rate, REM facilitates an increase of exchangeable nutrient supply and arthropod survival in soil. REM has a significant reduction of energy consumption and greenhouse gas emission over existing soil remediation practices. This study of interest and can be consider for publication after major revision.

Response: We appreciate the reviewer for the kind review and positive evaluation of our work. Point-by-point responses to the comments are provided below.

Comments:

1. In abstract, there is no mention of the scalability or cost-effectiveness of the REM process compared to conventional methods. Evaluating the scalability, economic feasibility, and practical implementation of this technique on a larger scale is necessary for its real-world applicability.

Response: We thank the reviewer for the comments. Following the reviewer's suggestion, we have added more descriptions of the comparisons between REM and other PFAS removal methods and revised the abstract as below.

We revised the abstract on P2.

“Per- and polyfluoroalkyl substances (PFAS) are persistent and bioaccumulative pollutants that can easily accumulate in soil, posing a threat to environment and human health. Current PFAS degradation processes often suffer from low efficiency, high energy and water consumption, or lack of generality. Here, we develop a rapid electrothermal mineralization

(REM) process to remediate PFAS-contaminated soil. With environmentally compatible biochar as the conductive additive, the soil temperature increases to >1000 °C within seconds by current pulse input, converting PFAS to calcium fluoride with inherent calcium compounds in soil. This process is applicable for remediating various PFAS contaminants in soil, with high removal efficiencies (>99%) and mineralization ratios (>90%). While retaining soil particle size, composition, water infiltration rate, and cation exchange capacity, REM facilitates an increase of exchangeable nutrient supply and arthropod survival in soil, rendering it superior to the time-consuming calcination approach that severely degrades soil properties. REM is scaled up to remediate soil at two kilograms per batch and promising for large-scale, on-site soil remediation. Life-cycle assessment and techno-economic analysis demonstrate REM as an environmentally friendly and economic process, with a significant reduction of energy consumption, greenhouse gas emission, water consumption, and operation cost, when compared to existing soil remediation practices.”

2. In introduction authors briefly discusses stabilization, chemical oxidation, and thermal treatment methods but lacks a detailed comparative analysis of their advantages, limitations, and efficiencies. Providing a more in-depth discussion and comparison of these methods would provide better context for why REM stands out. Take a look at this reference *Current Opinion in Chemical Engineering* 2023, 42:100954.

Response: We thank the reviewer for their suggestion regarding the detailed comparison between our REM method with other PFAS removal methods in the Introduction section. We revised the Introduction section in the manuscript accordingly on P4,

“The stabilization method involves mixing sorbents, such as activated carbon or clay, with the contaminated soil to sorb PFAS and reduce PFAS mobility and bioavailability¹¹⁻¹³. However, this method does not degrade PFAS in soil and sorbed PFAS could still pose long-term environmental damage. Chemical treatment uses strong oxidants to oxidize PFAS¹⁴⁻¹⁶. The residual oxidants need to be washed out with a large amount of water, and the wastewater could lead to secondary pollution to the environment. Traditional thermal treatment requires furnace heating for PFAS desorption and degradation, which often lasts for hours at 400-1100 °C¹⁷⁻²⁰. Some toxic short-chain fluorocarbon compounds, such as CF₄, C₂F₆, and C₂F₄, could be generated and emitted to the environment during this process. This is due to inadequate decomposition of C-F bonds, which will cause secondary pollution^{20,21}, and the extended heating also degrades soil properties²².”

References

11. Hale, S. E. et al. Sorbent amendment as a remediation strategy to reduce PFAS mobility and leaching in a contaminated sandy soil from a Norwegian firefighting training facility. *Chemosphere* 171, 9-18 (2017).
12. Das, P., Arias E, V. A., Kambala, V., Mallavarapu, M. & Naidu, R. Remediation of perfluorooctane sulfonate in contaminated soils by modified clay adsorbent—a risk-based

approach. *Water Air Soil Pollut.* 224, 1-14 (2013).

13. Kupryianchyk, D., Hale, S. E., Breedveld, G. D. & Cornelissen, G. Treatment of sites contaminated with perfluorinated compounds using biochar amendment. *Chemosphere* 142, 35-40 (2016).

14. Vecitis, C. D., Park, H., Cheng, J., Mader, B. T. & Hoffmann, M. R. Treatment technologies for aqueous perfluorooctanesulfonate (PFOS) and perfluorooctanoate (PFOA). *Front. Environ. Sci. Eng. China* 3, 129-151 (2009).

15. Dombrowski, P. M. et al. Technology review and evaluation of different chemical oxidation conditions on treatability of PFAS. *Remediation* 28, 135-150 (2018).

16. Mahinroosta, R. & Senevirathna, L. A review of the emerging treatment technologies for pfas contaminated soils. *J. Environ. Manage.* 255, 109896 (2020).

17. Endpoint. Bench-scale VEG research & development study: Implementation memorandum for ex-situ thermal desorption of perfluoroalkyl compounds (PFCs) in soils. (2017).

18. Söregård, M., Lindh, A. & Ahrens, L. Thermal desorption as a high removal remediation technique for soils contaminated with per-and polyfluoroalkyl substances (PFASs). *PloS one* 15, e0234476 (2020).

19. Xiao, F. et al. Thermal stability and decomposition of perfluoroalkyl substances on spent granular activated carbon. *Environ. Sci. Technol. Lett.* 7, 343-350 (2020).

20. Vargette, L. D. et al. Prospects of complete mineralization of per-and polyfluoroalkyl substances by thermal destruction methods. *Curr. Opin. Chem. Eng.* 42, 100954 (2023).

21. Alinezhad, A. et al. An investigation of thermal air degradation and pyrolysis of per- and polyfluoroalkyl substances and aqueous film-forming foams in soil. *ACS EST Engg.* 2, 198-209 (2022).

22. Zhao, C., Dong, Y., Feng, Y., Li, Y. & Dong, Y. Thermal desorption for remediation of contaminated soil: A review. *Chemosphere* 221, 841-855 (2019).

More detailed comparisons of these methods related to energy consumption, greenhouse gas emission, water consumption, and chemical waste generation have been listed in Fig. 5, Supplementary Note 4, Supplementary Table 11-17, and Fig. R1. Compared with other PFAS removal methods, REM exhibits 40% to 65% decrease in GHG emission (Fig. 5f and Supplementary Table 14), and 47% to 67% decrease in water consumption (Fig. R1a and Supplementary Table 15). Benefitting from the complete mineralization in the sealed system and no additional chemical usage, the REM method exhibits no additional chemical waste generation (Fig. 1b). REM can realize ~99.9% PFAS removal within seconds, achieving the best performance in overcoming the trade-off between removal efficiency and processing time among reported methods. The detailed comparisons of these methods are shown in Supplementary Fig. 65, which has been included in our first-version manuscript.

Fig. R1. Environmental impact comparisons between different PFAS removal methods. (a) Comparison of cumulative water consumption. (b) Comparison of chemical waste generation. (This figure has been added into SI as Supplementary Fig. 64)

3. While the manuscript briefly touches on the environmental impact, a more comprehensive discussion or comparison of the environmental impact of REM versus other methods would be beneficial. This could include not only energy consumption but also waste generation and the potential long-term effects on soil health and ecosystems.

Response: We thank the reviewer for the comment on the environmental impact of the REM method. In addition to energy consumption (Fig. 5e), we compared the greenhouse gas emission (GHG), water consumption, and chemical waste generation between our REM method with other PFAS removal methods (See details in Supplementary Note 4). Compared with other methods, such as thermal treatment, chemical oxidation, and ball milling, REM exhibits 40% to 65% decrease in GHG emission (Fig. 5f and Supplementary Table 14), and 47% to 67% decrease in water consumption (Fig. R1a and Supplementary Table 15). Benefitting from the complete mineralization in the sealed system and no additional chemical usage, the REM method exhibits no additional chemical waste generation (Fig. 1b). In addition, the REM method can remove >99% PFAS in seconds, which is several orders of magnitude faster than other methods (Supplementary Fig. 65). Therefore, REM shows superiorities with a much lower environmental impact.

Fig. R1. Environmental impact comparisons between different PFAS removal methods. (a) Comparison of cumulative water consumption. (b) Comparison of chemical waste generation. (This figure has been added into SI as Supplementary Fig. 64)

We have revised the manuscript, on P17-18,

“REM also exhibits 40% to 65% reduction in greenhouse gas emission (GHG, Fig. 5f), and 47% to 67% reduction in water consumption (Supplementary Fig. 64a) compared to other methods. The REM also has no chemical waste generation (Supplementary Fig. 64b).”

For the long-term effects of soil health, we first compared the particle size (Fig. 4b), specific area (Supplementary Fig. 44), soil composition (Fig. 4c,d), and water infiltration rates (Fig. 4e and Supplementary Fig. 45) of raw soil and REM soil, which show negligible differences, and were included in the first version of the manuscript. Here, we added the pH and cation exchange capacity (CEC) tests to the revised manuscript, where the pH and CEC values of raw soil and REM soil are also comparable (Table R1). Then, we tested the exchangeable nutrient content, including P, Ca, K, Mg, Mn, Fe, and N, in the soil and found that the contents of most exchangeable nutrients in REM soil increased by 10% to 102%, except for a ~5% decrease of Fe content (Fig. 4f, Supplementary Fig. 50). Finally, we cultured springtails and isopods in raw soil and REM soil, respectively, to evaluate the applicability of REM soil in the ecosystem. REM soil exhibited an arthropod survival ratio comparable to that in raw soil, indicating the good preservation of soil properties (Fig. 4g,h).

Table R1. Soil pH and CEC.

Soil types	pH	CEC (cmol kg ⁻¹)
Raw soil	7.19 ± 0.02	15.25 ± 0.81
REM soil	7.58 ± 0.03	15.45 ± 1.04

(This table has been included as Supplementary Table 6)

We have revised the manuscript, on P14,

“REM soil exhibits a pH of 7.58 that is slightly higher than that of raw soil (pH = 7.19) and the CEC is 15.45 cmol kg⁻¹, which is comparable to that of raw soil (15.25 cmol kg⁻¹, Supplementary Table 6).”

We have added soil CEC test procedures to the Methods section of the revised manuscript accordingly on P29-30,

“1.0 g of soil sample was dispersed into 10 mL of 1 M sodium acetate solution in an ultrasonic bath (Cole-Parmer Ultrasonic Cleaner) for 15 min to saturate soil exchange sites with Na⁺. Then, the sample was washed three times with ethanol (Decon’s Pure 200 Proof, Decon Labs Inc.) to remove the excess Na⁺. Afterwards, the sample was dispersed into 10 mL of 1 M ammonium acetate in an ultrasonic bath for another 15 min to replace Na⁺ by NH₄⁺ at exchange sites⁵⁵. The sample was then centrifuged (Adams analytical centrifuge, 60 rpm, 5 min), followed by filtration using a PES membrane (0.22 μm, Millipore-Sigma) to remove any undissolved particles. The filtrate was diluted to the appropriate concentration using 2 wt% HNO₃ within the ICP calibration curve range. The Na⁺ concentration was measured by inductively coupled plasma mass spectrometry (ICP-MS) using a Perkin Elmer Nexion 300 ICP-MS system, with sodium standard solution for ICP (1 g L⁻¹, Millipore Sigma) as the standard. Finally, CEC was calculated by equation (4),

$$CEC = \frac{c(\text{Na}^+) \times D \times V}{M(\text{Na}^+) \times m(\text{soil})} \quad (4)$$

where $c(\text{Na}^+)$ is the concentration of sodium measured by ICP-MS, D is the dilution factor, V is the volume of ammonium acetate solution to extract Na⁺ ($V = 10$ mL), $M(\text{Na}^+)$ is mole mass of sodium ($M(\text{Na}^+) = 23$ g mol⁻¹), $m(\text{soil})$ is the mass of soil sample used for CEC test ($m(\text{soil}) = 1.0$ g).”

Reference:

55. Carter, M. R. & Gregorich, E. G. *Soil sampling and methods of analysis*. (CRC press, 2007).

We have added soil pH test procedures in the Method section of the revised manuscript accordingly on P35,

“Soil pH was measured by a soil pH meter (SOILPHU), where the detector was inserted into 10 g of soil. The average pH values and the standard deviations for each sample were calculated after testing 5 times of.”

4. The manuscript mentions the successful remediation on a kilogram scale per batch but lacks discussion on the scalability and practicality of implementing REM on a larger, field-scale level. Further insights into the challenges, cost-effectiveness, and scalability for large-scale deployment would strengthen the manuscript.

Response: We appreciate the reviewer for the comment on the practicality of REM on field-scale soil remediation. To further demonstrate the scalability of REM at the field-scale level, we conducted an electric simulation of REM in a $1\text{ m} \times 1\text{ m} \times 1\text{ m}$ field. The voltage input of 1000 V is applied in this case (Fig. R2). The current density in the central position is calculated to be $\sim 800\text{ A m}^{-2}$, like that in the small-scale clay pot (Fig. 5b). According to our theoretical analysis (Supplementary Note 1), the current density determines the accessible temperature of the sample during the REM process. This indicates that a comparable temperature of $\sim 1000\text{ }^\circ\text{C}$ can be achieved under such a voltage input for the large-scale 1 m^3 sample. The current density is uniform both in-plane and in-depth (Fig. R2d,e), proving the homogeneous heating capability for field-scale soil remediation.

Our current work is a proof-of-concept work at a laboratory scale. There will be many challenges when further scaling the process up to a field-scale level, including the high voltage accessibility, the facility, and the equipment. These challenges would be addressed by the developer of the process since they are beyond our laboratory capabilities and the scope of this paper.

Fig. R2. Electric simulations at a $1\text{ m} \times 1\text{ m} \times 1\text{ m}$ field with 1000 V input. (a) Top view of the simulation geometry. (b) Lateral view of the geometry simulation. (c) Simulated potential distribution. (d) Simulated in-plane current density distribution. (e) Simulated cross-section current density distribution.

(This figure has been added into SI as Supplementary Fig. 59)

We revised the manuscript accordingly, on P16-17,

“The field-scale application potential of REM was evaluated by simulation at a $1\text{ m} \times 1\text{ m} \times 1\text{ m}$ scale (Supplementary Fig. 59). Under the voltage input at 1000 V, the current density of the 1 m^3 scale soil sample is like in the small-scale clay pot (Fig. 5b). Since the current density mainly determines the accessible temperature (Supplementary Note 1), we project a similar heating pattern can be achieved for the large-scale sample. In addition, the current density of the large-scale soil sample uniformly distributes both in-plane and in-depth (Supplementary Fig. 59d,e), confirming the homogenous heating capability of REM for on-site soil remediation.”

The discussion is also included in Supplementary Note 2,

“To assess the practical applicability of the REM process, we further extended the simulation to a $1\text{ m} \times 1\text{ m} \times 1\text{ m}$ scale. The configuration is shown in Supplementary Fig. 59a,b. The material properties and boundary conditions are the same as in the small-scale sample. In this case, the voltage input with 1000 V was applied (Supplementary Fig. 59c). The current density in the central position is calculated to be $\sim 800\text{ A m}^{-2}$, similar to that in the small-scale clay pot. According to our theoretical analysis (Supplementary Note 1), the current density determines the accessible temperature of the sample during the REM process. This indicates that a comparable temperature of $\sim 1000\text{ }^\circ\text{C}$ can be achieved under such a voltage input for the large-scale 1 m^3 sample.”

5. In Figures 1d and e, at 100 volts, the temperature during the first second of rapid thermal processing could potentially rise to 1300°C or even higher. However, the author discovered that the most effective mineralization occurred at 1000°C . Despite the expectation that higher temperatures might yield better mineralization, the author shall provide an explanation to the reader as to why this optimum temperature was chosen over even higher temperatures for enhanced mineralization.

Response: We thank the reviewer. First, according to our experimental data, the PFOA residue content monotonically decreased with the increase of input voltage (Fig. 1e and Fig. R3), which is consistent with our assumption that a higher temperature would facilitate better PFAS removal. However, the total F mass balance derived from the REM soil slightly decreased with the increase of input voltage from 100 V to 150 V (Fig. 1e). After removing soil samples from the quartz tube reactor, we found that there were some black residues adhering to the inner surface of the quartz tube (Fig. R4a) that were difficult to dissolve using water. We conducted XPS characterizations on the inner surface of the quartz tube and found that its F content increases with the increase of input voltage (Fig. R4b-f). This result indicates that higher amounts of insoluble F-containing compounds were deposited on the quartz tube with higher REM temperature, which influences the F^- mass balance, and thus

decreases the calculated mineralization ratio, especially at higher temperature.

Fig. R3. PFOA residue content with the input voltage increasing from 100 V to 150 V. The error bars denote standard deviations, where $N = 3$.

Fig. R4. F residue deposited on the inner surface of the quartz tube. (a) Picture of the inner surface of quartz tube after REM. (b-e) XPS spectra of the inner surface of the quartz tube after REM with the input voltage of (b) 80 V, (c) 100 V, (d) 120 V, and (e) 150 V. (f) F content on the inner surface of the quartz tube reacted under different input voltages. The error bars

in (f) denote standard deviations, where $N = 3$.

(Figure R4 has been added into the SI as Supplementary Fig. 14)

We revised the manuscript accordingly, on P8,

“The slight decrease of mineralization ratio and quantifiable total fluorine mass when the REM voltage increases from 100 V to 150 V (Fig. 1e) is attributed to the increased amount of insoluble F-containing compounds deposited on the quartz tube with the increase of REM temperature (Supplementary Fig. 14).”

6. Metcoke seem to appear as a more effective conductive additive when compared to biochar, demonstrating its advantage as approximately 91% by weight can be easily recycled after Rapid Thermal Processing (REM) through a simple sieving method. How author justify the use of biochar.

Response: We thank the reviewer for their comment on the choice of carbon additives. In this work, we demonstrated metcoke and biochar as two representative carbon additives for soil remediation. Each of them has their advantages.

(1) Metcoke: As the reviewer mentioned, metcoke is more conductive than biochar (Supplementary Table 2), and it is more easily recycled by a simple sieving method with a higher recycling yield of ~93 wt% (Supplementary Figs. 23-25) than that of biochar (~85 wt%, Supplementary Figs. 19-22). More importantly, the unit price of metcoke is ~\$150 tonne⁻¹, which is 1/10 of that of biochar (\$1400 tonne⁻¹). It indicates that the operating cost is much cheaper using metcoke as the carbon additive (\$30 tonne⁻¹), compared to using biochar as the carbon additive (\$130 tonne⁻¹, Supplementary Table 17).

(2) Biochar: Because of the relatively high Ca content of ~4 at% (Supplementary Fig. 32) in biochar than that in metcoke (lower than the XPS detecting limit, Supplementary Fig. 34), biochar can achieve a higher PFAS mineralization ratio (~94%) than that of metcoke (~91%, Supplementary Fig. 18). This indicates that biochar is a more favorable carbon additive for PFAS mineralization. In addition, biochar is well-known as a soil conditioner, which can facilitate plant growth when added to the soil by supplying more nutrients to soil (*Agron. Sustainable Dev.*, 2016, 36, 1-18). We extracted the exchangeable nutrients, including P, Ca, K, Mg, Mn, Fe, and N, in biochar and metcoke by Mehlich-3 reagent, respectively, and found that biochar has much higher nutrient content than that in metcoke (Fig. R5a). Considering the possible ion exchange during REM process, most exchangeable nutrient contents are higher in the biochar-assisted REM soil (denoted as B-REM soil) than those in the metcoke-assisted REM soil (denoted as M-REM soil), except iron (Fig. R5b). This indicates that when biochar is used as the carbon additive, the REM soil has a higher soil fertility, which would benefit soil biodiversity for real-world applications.

(3) Overall, we provide two possible options either using biochar or metcoke as the carbon additive for REM in this work. For real-world applications, the choice of carbon additives depends on the specific scenarios and requirements.

Fig. R5. Soil nutrient concentration measurements by different carbon additives. (a) Exchangeable nutrient contents of Ca, Fe, P, N, Mn, K, and Mg in the metcoke and biochar. (b) Exchangeable nutrient contents of REM soil using metcoke (M-REM soil) and biochar (B-REM soil) as the carbon additives. The error bars represent the standard deviation, where $N = 3$.

(This figure has been added into the SI as Supplementary Fig. 51)

We revised the manuscript accordingly, on P10, “For deployed applications, the choice of carbon additives depends on the specific scenarios and requirements”; and on P14-15, “We evaluated the influence of different carbon additives on soil nutrient contents and found that nutrient-rich biochar can facilitate higher exchangeable nutrient contents of REM soil, comparing with those of the REM soil using metcoke as carbon additive (Supplementary Fig. 51).”

7. In lines 176-180, the author should explain on how Ca^{2+} facilitates mineralization. They should delve into its chemistry and explain how it potentially impacts the reduction of C-F bond energy.

Response: We thank the reviewer for their comment on the role of Ca^{2+} in PFAS mineralization. To further determine the role of Ca in PFAS mineralization, we here added more simulations to explore the possible reaction pathway. In the simulation, four CaO pairs were placed on top of one PFOA molecule, and one red-arrow-pointed F atom was removed from PFOA with the help of Ca (Fig. R6a). The reaction barrier of the process is calculated to be 0.67 eV and the total energy is lowered by 1.24 eV (Fig. R6a), indicating that it is an energy-favorable reaction step. On the contrary, without Ca, the F atom spontaneously returned to its original position in the PFOA and reformed a covalent bond with the C atom, demonstrating the higher stability of the C-F bond.

In addition, the highly diffusive charge density of the highest occupied molecular orbital (HOMO) states (around the Fermi level at -2.6 eV) is localized in the CaO (Fig. R6b), indicating its high chemical reactivity before mineralization. In comparison, the Fermi level

of the final structure decreases to -4.8 eV (Fig. R5c), and the electron states in the F^- are lower than the HOMO states. This indicates that with the presence of Ca, F is more favorable to ionically bond with the Ca atom than forming a covalent bond with the C atom.

Fig. R6. Simulation results of PFOA mineralization by Ca. (a) Kinetic barrier of an F atom removal from a PFOA molecule with the help of Ca. Insets are the structures of the reactant, transition state, and product, and the red arrow marks the F atom being removed from the PFOA and then joining the CaO cluster during the process. (b) The initial structure and partial charge density of the CaO and PFOA molecules. (c) The final structure and partial charge density with the removal of one F atom from the PFOA. Colors of each element: red (O), brown (Ca), cyan (C), violet (F), and white (H).

(This figure has been added into the SI as Supplementary Fig. 37)

We have revised the manuscript accordingly, on P12,

“With Ca, F is more favorable to ionically bond with the Ca atom than forming a covalent bond with the C atom. The reaction barrier of C-F bond cleavage is calculated to be 0.67 eV and the total energy is lowered by 1.24 eV in the presence of Ca, indicating that the mineralization process is an energy-favorable reaction step. On the contrary, without Ca, the F atom spontaneously returned to its original position in the PFOA and reformed a covalent bond with the C atom. (Supplementary Fig. 37).”

8. Would other alkaline metals such as Mg or alkali metals potentially facilitate mineralization more effectively? The author should consider exploring this aspect as well.

Response: We appreciate the reviewer’s comment on the potential influence of other metal ions on PFAS mineralization. We here added more experiments by mixing different metal carbonates, including $CaCO_3$, $MgCO_3$, and Na_2CO_3 , as representative alkaline and alkaline earth metals, with PFOA. The metal counterion content is 1.2 mole equivalent compared to the F mole content in PFAS to ensure complete PFAS mineralization (Supplementary Table 4). After REM treatment, the initial PFOA peaks in XRD patterns vanished, and the metal fluoride peaks appeared (Fig. R7), which indicates that all these alkaline and alkaline earth metal ions can be used for PFAS mineralization.

Fig. R7. XRD patterns of different metal carbonates mixed with PFAS before (red line) and after (blue line) REM. (a) CaCO_3 . (b) MgCO_3 . (c), Na_2CO_3 . The PDF reference cards for each are CaCO_3 , 01-085-0849; CaO , 00-048-1467; CaF_2 , 04-008-4867; MgCO_3 , 00-025-0513; MgO , 01-076-2583; MgF_2 , 00-041-1443; Na_2CO_3 , 00-037-0451; NaF , 01-080-8614.

(This figure has been added into the SI as Supplementary Fig. 28)

After further measuring the residual PFOA content in the products by LC-MS, it was found that CaCO_3 can reach a higher PFOA removal efficiency (~99.7%) than MgCO_3 (~94.2%) and Na_2CO_3 (~98.5%, Fig. R8), proving Ca has the best PFAS mineralization performance, greater than Mg and Na.

Fig. R8. PFOA removal efficiency by mixing different metal carbonates with PFOA for REM. The error bars denote standard deviations, where $N = 3$.

(This figure has been added into the SI as Supplementary Fig. 29)

To explain the difference, we compared the bond energy between F and different alkaline and alkaline earth metal ions present in soil (Table R2), where Ca-F has the highest bond energy. The result indicates that Ca is more likely to mineralize PFAS than other metal cations. In addition, once other metal fluorides like MgF_2 or NaF are formed, it is thermodynamically favorable for them to convert back to CaF_2 during REM with a negative ΔG under a temperature higher than 400 °C (Fig. R9), since Ca content in soil (4-5 at%, Supplementary Figs. 40 and 41) is >100 times more concentrated than PFAS in soil (~100 ppm) in some areas with the highest contents of contaminations (*Sci. Total Environ.* 2020, 740, 140017). To conclude, we do not exclude the effect of other alkaline and alkali metals,

but the above analysis shows that Ca is the best metal for the mineralization of PFOA.

Table R2. Metal-fluorine bond energy.

M-F type	Bond energy (kJ mol ⁻¹)
Na-F	477
Mg-F	463
K-F	489
Ca-F	529

(This table has been added into the SI as Supplementary table 4)

Fig. R9. The Gibbs free energy change (ΔG) of the conversion from other metal fluorides to CaF_2 under different temperatures.

(This figure has been added into the SI as Supplementary Fig. 30)

We have revised the manuscript accordingly, on P10-11,

“To confirm the influence of Ca on PFAS mineralization, we first compared the mineralization performance of Ca^{2+} with other alkali and alkaline earth metal ions, such as Mg^{2+} and Na^+ , where calcium carbonate (CaCO_3 , a representative calcium specie in soil, magnesium carbonate (MgCO_3) and sodium carbonate (Na_2CO_3), were separately mixed with PFOA and the metal counterion content is 1.2 mole equivalent compared with F (Supplementary Table 4). After REM treatment, X-ray diffraction (XRD) patterns show the loss of PFOA peaks and the appearance of metal fluoride peaks (Supplementary Figs. 27 and 28), indicating that all these alkaline and alkaline earth metal ions can be used for PFAS mineralization. However, Ca achieved a highest PFOA removal efficiency ($\sim 99.7\%$) over Mg ($\sim 94.2\%$) and Na ($\sim 98.5\%$, Supplementary Fig. 29), proving that Ca has the best mineralization performance for PFAS, greater than Mg and Na. In addition, the Ca-F bond has the highest bond energy among different metal-fluorine bonds (Supplementary Table 5). Theoretical calculation reveals that once other metal fluorides like MgF_2 or NaF are formed,

it is thermodynamically favorable for them convert to CaF_2 during REM under a temperature higher than $400\text{ }^\circ\text{C}$ (Supplementary Fig. 30). The above analysis shows that Ca is the best metal for the PFAS mineralization process.”

We also supplemented the experimental details in the revised Method section, on P21, “To investigate the cation influence on the PFAS mineralization, different alkaline and alkaline earth carbonates, including CaCO_3 , MgCO_3 , and Na_2CO_3 , were mixed with PFAS. The metal counterion content is 1.2 mole equivalent compared with the F mole content in PFAS to ensure complete PFAS mineralization. Metcoke was used as the carbon additive and the total sample mass per batch was set as 300 mg. During REM, the input voltage was set as 100 V, and the discharging time was set as 1 s (See details in Supplementary Table 4).”

9.Has the author identified the formation of any side short-chain degradation products in the absence of Ca?

Response: We thank the reviewer for this comment. To identify the products in the absence of Ca, we captured the gaseous byproduct during REM process by replacing soil with SiO_2 , which is the main component of soil but without Ca, using Ca-free metcoke as the carbon additive. After analyzing the evolved gas by GC-MS, some fluorinated compounds, such as CF_4 , CH_3F , C_2F_6 , C_2F_4 , and $\text{C}_6\text{H}_5\text{F}$, were observed (Fig. R10), which are consistent with the reported PFOA-degradation products (*Nature*, 2001, 412, 321-324; *Anal. Chem.*, 2004, 76, 3800-3803.). In contrast, none of these products was found in the evolved gas from soil containing Ca (Supplementary Fig. 12). This underscores the critical role of Ca in soil for PFAS mineralization by REM, avoiding the emission of PFAS-degraded short-chain fluorocarbon species.

Fig. R10. GC-MS results of the evolved gas during the REM process. (a) GC-MS chromatogram of the gases from the mixture of SiO₂ and PFOA. (b-f) Zoom-in GC-MS chromatograms with different retention times at (b) 1.875 min, (c) 1.963 min, (d) 2.149 min, (e) 2.581 min, (f) 3.416 min.

(This figure has been added into the SI as Supplementary Fig. 13)

We have revised the manuscript accordingly, on P8,

“Some PFOA-degraded fluorinated compounds, such as CF₄, CH₃F, C₂F₆, C₂F₄, and C₆H₅F, were observed when replacing soil with SiO₂ in the absence of Ca (Supplementary Fig. 13). This indicates that REM in the presence of Ca effectively mineralizes F from soil contaminated with PFAS and avoids the emission of PFAS degraded short-chain fluorocarbon species.”

10. How does the author elaborate on the setup's capacity to effectively mineralize PFAS-contaminated soil—up to 99%—and its correlation with the electrodes' surface area in terms of the amount (in kilograms) processed by the current setup??

Response: We appreciate the reviewer's question on the relationship between electrode surface area and PFAS mineralization efficiency. First, we conducted REM experiments in the tubes with different inner diameters of 4 mm, 8 mm, and 16 mm, where the surface area of each electrode can be calculated from the cross-section area. We kept the same energy density input during REM and tested the residue PFOA content in different tube reactors.

Then, we defined r as the specific area ratio of the electrode (unit: cm² g⁻¹) by $r = \frac{2S}{m}$, where m is the REM sample mass and S is the surface area of each electrode. In the quartz tube reactor, 2 electrodes were placed on each side of the tube. The relationship between PFOA removal efficiency with r is shown in Fig. R11, which indicates that a higher surface area of electrodes can better facilitate PFOA removal.

Fig. R11. PFOA removal efficiency with different specific area ratios. The error bars denote standard deviations, where $N = 3$.

To further explore the relationship between PFAS removal efficiency with electrode surface area, we simulated the current density distribution on a $1\text{ m} \times 1\text{ m} \times 1\text{ m}$ sample volume with different electrode areas (Fig. R12). With the increase of each electrode surface area from 0.25 to 0.75 m^2 , the current density at the center position increases from 730 to 870 A m^{-2} (Fig. R13), which indicates a higher REM temperature since the current density determines the temperature according to our analysis (Supplementary Note 1). Therefore, during REM, a higher electrode surface area can facilitate a higher REM temperature and thus a higher PFOA removal efficiency. This is consistent with our experimental results (Fig. R11).

Fig. R12. Simulated current density distribution with different electrode surface areas. (a-c) Geometry and boundary conditions. (d-f) Current density distribution.

(This figure has been added into SI as Supplementary Fig. 60)

Fig. R13. Current density at the center position versus different electrode surface area.
(This figure has been added into SI as Supplementary Fig. 61)

We have revised the manuscript accordingly, on P17.

“Based on the simulation results, the increase of electrode surface areas facilitates an increase of current density with a certain voltage input, leading to a higher REM temperature for PFAS mineralization (Supplementary Figs. 60 and 61).”

The discussion is included in Supplementary Note 2.

“To reveal the relationship of electrode surface area on the heating efficacy, we conducted a simulation on a 1 m × 1 m × 1 m volume sample with different electrode surface area (Supplementary Fig. 60). With the increase of each electrode surface area from 0.25 to 0.75 m², the current density at the center position increases from 730 to 870 A m² (Supplementary Fig. 61), which indicates a higher REM temperature since the current density determines the temperature according to our analysis (Supplementary Note 1). Therefore, during REM, a higher electrode surface area can facilitate a higher REM temperature and thus a higher PFOA removal efficiency.”

11. What would be the optimal ratio of biochar or metcoke in relation to a specified quantity of soil? How it normally investigate.

Response: We thank the reviewer for the question on the optimal ratio of carbon additive and soil. The ratio of soil to the carbon additive is related to the sample resistance and thus influences the temperature during the REM process with a certain voltage input. Experimentally, we changed the soil:biochar and soil:metcoke mass ratios from 1:1 to 1:4. When the ratio of carbon additive is decreased, the sample resistance increases, leading to a decrease of the REM temperature with the same input voltage of 100 V (Fig. R14a,b). Thus, the PFOA removal efficiency decreases (Fig. R14c,d). However, considering the inferior conductivity of biochar, the sample resistance and REM temperature changes more than when using metcoke as the carbon additive (Fig. R14a). The high PFOA removal efficiency

(>99%) can only be achieved with a relatively low soil:biochar ratio of no more than 2:1 (Fig. R14c). The PFOA removal efficiency remained at a high value of >96% when changing the soil:metcoke ratio from 1:1 to 4:1 (Fig. 14d). Therefore, during REM process, the mass ratio of soil and carbon additive has been set as 2:1 to balance the tradeoff between PFOA removal efficiency and consumption of carbon additives.

Fig. R14. PFOA removal with content of carbon additives by REM. (a) Sample resistance and REM temperature with different soil/biochar ratio. (b) Sample resistance and REM temperature with different soil:metcoke ratio. (c) PFOA removal efficiency with different soil:biochar ratio. (d) PFOA removal efficiency with different soil:metcoke ratio. The error bars in (c) and (d) denote standard deviations, where $N = 3$.

(This figure has been added into the SI as Supplementary Fig. 26)

We have revised the manuscript accordingly, on P10.

“The optimal ratio between soil and different carbon additives was also investigated, where sufficient carbon additive content (>33 wt%) is required to ensure REM temperature for PFAS mineralization (Supplementary Fig. 26).”

12. Line 289 How author calculate the energy consumption of the REM process is calculated to be $\sim 420 \text{ kWh ton}^{-1}$.

Response: We thank the reviewer for the comment on the energy consumption of that REM process. We kindly remind the reviewer that the calculation details were shown in Supplementary Note 3 in the first version of the manuscript. Briefly, during our REM process, 200 mg of soil was mixed with 100 mg of biochar per batch. The input voltage was set as 100 V and the REM system capacitance was 60 mF. When flashed once to remediate

200 mg of soil, the energy consumption can be calculated as follows: $E = \frac{(V_1^2 - V_2^2) \times C \times n}{2 \times M}$

Where E is the consumed energy per gram (kJ g^{-1}), V_1 and V_2 are the voltage before and after REM, respectively, C is the capacitance ($C = 60 \text{ mF}$), n is the times of pulses, and M is the mass per batch. The energy consumption can be thus calculated to be 1.5 kJ g^{-1} or 420 kWh t^{-1} .

Reviewer #2 (Remarks to the Author):

In this work, the authors reported a rapid electrothermal mineralization (REM) process to remediate PFAS-contaminated soil. PFAS are known as hard-to-degrade chemicals. Current methods make it difficult to achieve high removal efficiency and defluorination efficiency simultaneously. In this study, the author utilized the REM process to achieve a good PFAS removal effect and defluorination effect. The superiority of REM over various kinds of PFAS removal methods was presented, and the reaction mechanisms were thoroughly discussed. It furnishes us with new ideas for dealing with environmental PFAS in future work. However, there are a few issues that need to be addressed before I recommend its publication in Nature Communications. The specific comments are as follows.

Response: We appreciate the reviewer for the positive evaluation to our work. Point-by-point responses to the reviewer' comments are shown below.

1. In line 122, the authors mentioned: “two O-rings on each electrode to seal the reacting tube during REM”. To confirm the necessity of the sealed system, the authors are supposed to supplement more experiments to show the trends of PFOA mineralization efficiency varied with the input voltage in the open system without an O-ring.

Response: We thank the reviewer for their comment on the PFOA mineralization efficiency trends with the increase of the input voltage in an open system. We here conducted REM experiments in an open system without the O-rings to seal the quartz tube (Fig. R15a) and tested the mineralized F⁻ and residue PFOA content in the REM soil (Fig. R15b). With the increase of input voltage, the PFOA content decreases, benefitting from a higher reaction temperature. However, the total fluorine content significantly decreases with the increase of input voltage, with only half of the organic fluorine mineralized into fluorine ions, which can be ascribed to the emission of PFOA-degraded short-chain perfluorinated species.

Fig. R15. PFOA mineralization in an open system. (a) Picture of the REM reaction jig without O-ring sealing. (b) Concentrations of organic fluorine and mineralized fluorine ion in PFOA-contaminated soil varied with input voltages, conducted in an open REM system

without O-rings to seal the quartz tube. The error bars represent the standard deviation, where $N = 3$.

(This figure has been added into SI as Supplementary Fig. 8)

We have revised the manuscript accordingly, on P7,

“REM was initially conducted in an open system without O-rings to seal the quartz tube. With the increase of input voltage, the PFOA content decreases, benefitting from a higher reaction temperature (Supplementary Fig. 8). However, the total fluorine content significantly decreases with the increase of input voltage, with only half of the organic fluorine mineralized into fluorine ions, which can be ascribed to the emission of PFOA-degraded short-chain species (Supplementary Fig. 9).”

2. In Figure 4, the soil properties between raw soil, REM soil, and calcined soil were compared. However, the residual PFOA content and F mineralization content in calcined soil should be supplemented to avoid interference from residue PFOA.

Response: We appreciate the reviewer’s comment on the PFOA and F mineralization content in the calcined soil. We tested the residual PFOA and F^- content in calcined soil and compared them with those in REM soil. After 2 h calcination at 900 °C, the PFOA removal efficiency reached a value of >99.9% (Fig. R16a), which is comparable with that of REM process. However, the F mineralization efficiency in the calcined soil is only ~0.34% (Fig. R16b), which is much lower than the F mineralization efficiency in REM soil (~94%). Considering the exposure to an open environment during the calcination process, the results indicate that PFOA volatilized or it degraded into toxic short-chain perfluorinated species, which were emitted into the atmosphere and led to secondary pollution.

Fig. R16. Comparison of PFOA mineralization performance between REM and calcination. (a) PFOA removal efficiency. (b) PFOA mineralization ratio. The error bars in (a) and (b) denote standard deviations, where $N = 3$.

(This figure has been added into SI as Supplementary Fig. 10)

We have revised the manuscript accordingly, on P8,

“By virtue of the sealing design, REM soil shows a much higher mineralization ratio (94%) compared with the furnace-calcined soil (0.34%), while keeping a high and comparable

PFOA removal efficiency of >99% (Supplementary Fig. 10).”

3. In the Methods section, the soil was dried before REM, but most of the soil in nature contains a certain amount of moisture. Can PFOA in wet soil also be effectively removed by the REM method? In other words, is pre-drying a required step for the contaminated soil before the REM process?

Response: We appreciate the reviewer’s comments on the applicability of the REM process for PFAS mineralization in moisture-containing soil. We measured the moisture content of the soil used in our experiments. The previously used dry soil had a moisture content of ~1.1 wt% (Figure R17a). Here, we collected another batch of soil with a moisture content of ~30 wt% (denoted as wet soil, Figure R17b). We mixed PFOA with the wet soil and used biochar as the conductive additive. The input voltage of the REM was kept the same as that for the dry soil (Supplementary Table 2). The content of residual PFOA progressively decreased with increasing electric pulses and was reduced to below the residential soil remediation standards after 2 pulses (Figure R17c-d), like the results obtained from the dry soil (Figs. 1e). The F mineralization ratio of the wet soil after REM reaches 93%, which is comparable to that of the dry soil (94%). This demonstrates the feasibility of the REM process to mineralize PFAS in the wet soil with a moisture content of ~30 wt%. Pre-drying is not a required step for such moisture level.

Figure R17. PFOA mineralization in wet soil. (a) TGA curve of dry soil. (b) TGA curve of wet soil. TGA is conducted in air with a heating rate of 5 °C min⁻¹ and then kept at 110 °C for 30 min. (c) Residual PFOA concentrations in soil after repetitive electric pulses, with an input voltage of 100 V and duration of 1 s each time. (d) Comparison of mineralization ratios in dry soil and wet soil. The error bars in c and d denote standard deviations, where $N = 3$. (This figure has been added into SI as Supplementary Fig. 62)

We have revised the manuscript accordingly, on P17,

“For the practical application, considering the moisture contained in field soil, we assessed the applicability of REM for PFAS mineralization in wet soil. After REM, the wet soil with a moisture content of ~30 wt% achieved a PFOA mineralization ratio comparable to pre-dried soil (Supplementary Fig. 62), further suggesting the feasibility of REM for practical deployment.”

4. In the Methods part, the authors mentioned a filtration process using a PES filter before further LC-MS tests. I wonder if the membrane can absorb PFAS during filtration. More experiments are needed to compare the PFAS content in the supernatant and filtered leaching solution.

Response: We appreciate the reviewer’s perceptive comment on the PES filter in PFAS measurement. To investigate the influence of the PES filter on PFOA measurement, we extracted PFOA from soil, centrifugated, and then tested PFOA content in the supernatant and the filtered solution by LC-MS. No obvious differences in PFOA content were observed in the supernatant and the filtered solution (Fig. R18), indicating that the filter does not influence the PFOA content measurement.

Fig. R18. Boxplot with individual data points of PFOA contents extracted from soil with and without the PES filter. $N = 6$. The central line represents the median value. Box limits represent upper and lower quartiles. Whiskers represent the 5th and 95th percentiles.

(This figure has been added into SI as Supplementary Fig. 66)

We have revised the Methods section of the manuscript accordingly, on P22,

“The PES filter had a negligible influence on PFAS detection (Supplementary Fig. 66).”

5. The authors just listed the temperature profiles in Figure 1d. More temperature-time curves for different voltages, like 80 V, 120 V, and 150 V should be provided.

Response: We appreciate the reviewer for this suggestion. The temperature-time curves with

the input voltage of 40 V, 60 V, 80 V, 120 V and 150 V are provided in Fig. R19. A higher voltage input leads to a higher temperature and the relationship between the temperature and input voltage is shown in Supplementary Fig. 3. Note that with the input voltage of < 100 V, the REM peak temperature is lower than 1500 °C. Therefore, the temperature was recorded using the IR thermometer (Micro-Epsilon) with a detection range of 200 to 1500 °C. In contrast, with the input voltage of 120 V and 150 V, the REM temperature was recorded using another IR thermometer (Micro-Epsilon) with a detection range of 1000 to 3000 °C.

Figure R19. Temperature measurements with different voltage inputs. (a) 40 V, (b) 60 V, (c) 80 V, (d) 120 V, and (e) 150 V.

(This figure has been included in Supplementary Fig. 2)

6. In Supplementary Fig. 31, the authors mentioned “the particles with size < 2 μm are cataloged as clay, while the size in the range of 2-50 μm for silt, and the size > 50 μm for sand.” More references here need to be provided for the soil classifications.

Response: We thank the reviewer for the comment on soil classifications. We added more references on P13.

“Laser particle size analysis results also reveal comparable size distributions between raw soil and REM soil, but a significant increase of particle sizes with much lower clay and silt ratio^{41,42} after calcination.”

References

41. Faé, G. S., Montes, F., Bazilevskaya, E., Añó, R. M. & Kemanian, A. R. Making soil particle size analysis by laser diffraction compatible with standard soil texture determination methods. *Soil Sci. Soc. Am. J.* 83, 1244-1252 (2019).

42. Barman, U. & Choudhury, R. D. Soil texture classification using multi class support vector machine. *Inf. Process. Agric.* 7, 318-332 (2020).

7. There are some spelling mistakes in the paper. For example, Line 101, "Universtisy" should be corrected to "University". please check the entire text.

Response: We appreciate the reviewer pointing out our spelling mistake. We have changed the "University" in P6 and checked the entire document.

Reviewer #3 (Remarks to the Author):

This is a very interesting communication. The study describes how soil is cleaned up from PFAS by electrothermal heating. With low energy and very fast the fluorine from PFAS can be mineralised into benign calcium fluoride by not changing the soil properties significantly. The authors made an astonishing multi-platform analytical effort to show that not only PFAS was reduced to fluoride but that the soil properties did not change significantly. The method is well described and the different analytical methodologies used have sufficient information to redo the experiments - hence transparent enough.

This is the first time that I have seen results of a clean up method for PFAS which can actually work on a larger scale. Hence, that this study could have a significant effect on environmental science. Hence, I would recommend to publish this study in Nature Comm. after a successful rebuttal of those queries.

Response: We appreciate the reviewer for the positive evaluation of our work. Point-by-point responses to the reviewers' comments are shown below.

1. There are however some queries to be answered. The effectiveness was mainly studied first with PFOA and other compounds, but polymers were not so much featured in the study (although PTFE was shown to be degraded in soil by IR and XRD).

Response: We thank the reviewer for their question on PTFE mineralization during the REM process. First, we tested the total fluorine content by combustion ion chromatography (CIC) before and after REM. No obvious difference was observed after REM (79.9 ppm for initial soil, and 78.1 ppm for REM soil), which indicates negligible emission of gaseous perfluorinated species (Fig. R20). Then, mineralized F⁻ contents in soil samples with different input voltages were further tested by IC. The PTFE mineralization ratio was calculated by dividing the initial CIC-tested F content by IC-tested F⁻ content in REM soil. The F⁻ contents after REM increase with an increase of input voltage from 0 to 150 V and an optimal mineralization ratio of 95% was obtained with an input voltage of 150 V (Fig. R21). It

indicates that the REM method works for both short-chain PFAS and F-contained polymers, such as PTFE.

Fig. R20. Total F content for PTFE-contaminated soil before and after REM. The REM was conducted with an input voltage of 100 V with a duration time of 1 s. The fluorine contents were tested by the CIC methods and the error bars denote standard deviations, where $N = 3$. (This figure has been included as a part of Supplementary Fig. 15)

Fig. R21. PTFE mineralization ratios vary with different input voltages. The error bars denote standard deviations, where $N = 3$. (This figure has been included as a part of Supplementary Fig. 15)

We have revised the manuscript accordingly, on P9,

“Beside short-chain PFAS, REM is also applicable to mineralize F-containing polymers, such as polytetrafluoroethylene (PTFE) with a high mineralization ratio of ~95% (Supplementary Fig. 15).”

We added experimental details for CIC tests in the revised Methods section, on P23,

“The soil sample (~10 mg) was loaded into a combustion furnace (AQF-2100H, NITTOSEIKO ANALYTECH) with the temperature of 1100 °C under 400 mL min⁻¹ oxygen

flow. The combusted anions were absorbed by 100 mL min⁻¹ water-saturable Ar and 200 mL min⁻¹ Ar, and then flowed into a gas absorption unit (GA 211, Mitsubishi Chemical Analytech). Afterwards, total F content was analyzed by an IC system (Dionex ICS-2100, Thermo Scientific).”

We also added calculation procedures for PTFE mineralization ratios, on P24, “The mineralization ratio (*R*) of PFAS is calculated according to equation (2),

$$R = \frac{c(\text{F-ion}) \times D_1}{c(\text{PFAS}) \times r \times D_2} \times 100\% \quad (2)$$

where *c*(PFAS) is the concentration of PFAS measured by LC-MS, *c*(F-ion) is the concentration of fluorine ions measured by IC, *r* is the mass ratio of fluorine atom in a certain PFAS molecular (listed in Supplementary Table 1), and *D*₁ and *D*₂ are the dilution factors of PFAS and fluorine ions, respectively. Note for the PTFE, *c*(PTFE) was calculated by dividing initial F content from CIC data by *r*(PTFE).”

2. The authors should however also give for each method especially IR, 19F-NMR and XRD the limit of detection in absolute amount as of in ng/g in order to better interpret the graphs and please give also the numbers in a table. This would help to interpret the analytical results. Are these values appropriate for environmental applications? could we detect relevant concentrations?? and can we reduce the PFAS concentration to low enough concentrations for todays regulations??

Response: We thank the reviewer for the comment on PFAS detection limits of different kinds of characterization methods. We list the detection limits of 5 kinds of characterization methods used in this work in Table R3. In this paper, we applied HPLC-DAD and QQQ LC-MS to quantify the PFAS content before and after REM because of its low detection limit. In contrast, other methods, such as FT-IR, XRD, and ¹⁹F-NMR, were used to qualitatively confirm the PFAS removal and mineralization after REM.

Table R3. Detecting limit of different PFAS characterization methods.

Characterization methods	Detecting limit
FT-IR	1wt% (ref ²⁷)
XRD	0.5 wt% (ref ²⁸)
¹⁹ F-NMR	50 ppb (ref ²⁹)
HPLC-DAD	500 ppb
QQQ LC-MS	0.1 ppb

(This table has been included in Supplementary Table 3)

Supplementary References:

27. Gorrochategui, E., Lacorte, S., Tauler, R. & Martin, F. L. Perfluoroalkylated substance effects in *Xenopus laevis* A6 kidney epithelial cells determined by ATR-FTIR spectroscopy and chemometric analysis. *Chem. Res. Toxicol.* 29, 924-932 (2016).

28. Hillier, S. Quantitative analysis of clay and other minerals in sandstones by X-Ray

powder diffraction (XRPD). *Clay mineral cements in sandstones* 213-251 (1999).

29. Camdzic, D. et al. Quantitation of total PFAS including trifluoroacetic acid with fluorine nuclear magnetic resonance spectroscopy. *Anal. Chem.* 95, 5484-5488 (2023).

Based on the LC-MS data, we found that PFOA content can significantly decrease below the residential soil remediation standards (130 ppb, the New Jersey Department of Environmental Protection) after 2 electric pulses and can be further reduced to 1.1 ppb after 4 electric pulses (Fig. 1f), which indicates that PFAS residue content in REM soil can meet the soil regulations. We also demonstrate the arthropod culture in the REM soil, which exhibits a comparable arthropod survival ratio with that of raw soil (Fig. 4g,h), supporting the environmental applicability for REM soil.

3. I do not have experience in XRD, but I do not understand that all other minerals in the soil (and I guess that they are not all amorphous) can be cancelled out to show only the PFAS?? Here a better explanation should be given.

Response: We appreciate the reviewer's comment on XRD characterizations of the PFAS mineralization products. Considering the PFAS content in soil is usually lower than 200 ppm (*Sci. Total Environ.* 2020, 740, 140017), the content of its mineralized products, such as CaF₂, is lower than the detection limits for XRD. Other mineral compounds in soil would also influence the detection of CaF₂ by XRD. Therefore, to confirm the influence of Ca²⁺ on PFAS mineralization, we did not use soil to mineralize PFAS. Instead, we chose CaCO₃, a representative calcium species in soil, as the mineralization agent, directly mixed it with PFAS, and kept a Ca/F ratio of 0.6 to ensure 20 at% excess of Ca (Supplementary Table 4). After the REM process, the obvious CaF₂ in XRD patterns can be observed for different kinds of PFAS (Supplementary Fig. 31), demonstrating the critical role of Ca²⁺ in PFAS mineralization. To avoid misunderstanding, we clearly stated in the manuscript that the XRD patterns were collected on the PFAS/CaCO₃ sample, instead of soil samples.

4. Maybe I missed it but what I need to see is a CIC analysis given the EOF which is often used to identify all extractable organofluorine in a sample. This would be rather interesting especially for PTFE in soil.

Response: We thank the reviewer for their suggestion on CIC test of the total F content. We agree that CIC is a useful tool to quantify the total F content in soil, and thus we here supplemented more CIC tests. We chose PFOA and PTFE as two representative PFAS for the CIC test before and after REM. According to the CIC data, no obvious changes of total F contents were observed for both PFOA and PTFE after REM (Fig. R22), which confirms that in our sealed REM system, negligible gaseous perfluorinated species were evolved. In addition, based on the CIC data, we also calculated the mineralization ratios of PTFE with different voltage inputs, as shown in Fig. R21 for Question 1 of this reviewer.

Fig. R22. Total F content before and after REM tested by CIC. (a) PFOA soil. (b) PTFE soil. The REM was conducted with an input voltage of 100 V with a duration of 1 s. The error bars denote standard deviations, where $N = 3$.

(This figure has been included as a part of Supplementary Fig. 15)

5. The LC-MS method is not appropriate in all cases here. It should be an LC-MS/MS method or even better an LC-HRMS for non-targeted analysis to identify all PFAS and not only the target methods. The reviewer would like to see an LC-HRMS before and after a F-containing polymer has been degraded.

Response: We thank the reviewer for their comment on the F-containing polymer degradation process. The LC-MS equipment we used is QQQ LC-MS with an LC-MS/MS method, instead of the conventional LC-MS method, where we can analyze 24 different kinds of PFAS at the same time. We are sorry for the misunderstanding and have supplied more detailed descriptions in the revised Methods section, on P22.

“In order to detect the trace amount of residual PFOA after REM (<1 ppm), a triple quadrupole (QQQ) LC/MS system (6740B LC/TQ, Agilent) using dynamic multiple reaction monitoring (DMRM) was applied. Here, the chromatographic separation was performed on a C18 analytical column (Zorbax Eclipse Pluse C18 Rapid Resolution HT, 2.1×50 mm 1.8-micron column, Agilent) with an ultra-high-performance LC (UHPLC) system (1290 Infinity II, Agilent). The aqueous phase consisted of 20 mM ammonium acetate solution, and the organic phase of methanol. The column was operated at a temperature of 40 °C and 40 μ L sample was injected each time. The mobile phase flow rate was maintained at 0.4 mL min^{-1} throughout the run. The column was equilibrated at initial conditions for 3 min before the next injection. The LC-MS system was interfaced to the MS system through an Agilent Jet Spray (AJS) electrospray ionization (ESI) source that was operated in the negative ionization mode.”

For an F-containing polymer like PTFE, its degradation products are fluorocarbon with no functional groups for proton donation and accepting, which makes it inaccessible to investigate PTFE degradation products by our LC-MS/MS method. Instead, we first

detected the gaseous degradation products of PTFE by collecting evolved gas. Trace amounts of tetrafluoroethylene (C_2F_4) and trifluoromethanol (CF_3OH) were detected with a retention time of 1.884 min and 2.591 min (Fig. R23), which are consistent with previously reported PTFE degradation products (*Mater. Chem. Phys.*, 2019, 221, 436-446; *Russ. J. Phys. Chem. A*, 2020, 94, 2135-2140). Then, we extracted the possible PTFE degradation compounds in the REM soil using a mixture solution of methanol, acetone, and toluene. Then, the extractant was detected by GC-MS. The main peak at the retention time of 28.551 min can be ascribed to siloxane, while no fluorinated species were detected in the REM soil (Fig. R24).

Fig. R23. GC-MS results of evolved gas of PTFE-contaminated soil during REM. (a) GC-MS chromatogram of the gases from PTFE-contaminated soil. (b-c) Zoom-in GC-MS chromatograms with different retention times. (b) 1.884 min; (c) 2.591 min.

(This figure has been included as a part of Supplementary Fig. 16)

Fig. R24. GC-MS results of the solvent-extracted phase of REM soil. (a) GC-MS chromatogram of the gases from PTFE-contaminated soil. (b-d) Zoom-in GC-MS chromatograms with different retention times. (b) 7.648 min; (c) 10.949 min; (d) 28.551 min. (This figure has been included as a part of Supplementary Fig. 17)

We have revised the manuscript accordingly, on P9,

“In addition to short-chain PFAS, REM is also applicable to mineralize F-containing polymers, such as polytetrafluoroethylene (PTFE) with a high mineralization ratio of ~95% (Supplementary Fig. 15). Trace amounts of PTFE degradation compounds, including tetrafluoroethylene and trifluoromethanol were detected in the gaseous phase during REM (Supplementary Fig. 16), while none of fluorinated compounds were detected in the REM soil (Supplementary Fig. 17).”

We also supplemented experimental details for GC-MS tests in the revised Methods section, on P25,

“For the gas detection, the injector and the transfer line were set with the temperature of 120 and 200 °C, respectively. The temperature program was initiated at 48 °C for 3 min, and then increased to 80 °C at 8 °C min⁻¹. The carrier gas was helium at a flow rate of 0.5 mL min⁻¹. For the F-containing residue detection, ~200 mg REM treated soil samples were added into 5 mL extractant solvent (mixture of hexane, acetone, and toluene with volume ratio of 10:5:1). Then, the mixture was immersed into an ultrasonic bath (Cole-Parmer Ultrasonic Cleaner) for 15 min for the extraction, followed by centrifugation (Adams Analytical Centrifuge, 60 rpm) for 2 min, and filtration using PES membrane (0.22 µm, Millipore-Sigma) to remove any undissolved particles. The filtered solution was loaded onto a GC autosampler. During the test, the injector and the transfer line temperature were set to 280 and 300 °C, respectively. The temperature program was initiated at 75 °C for 1 min, increased to 230 °C at 10 °C min⁻¹ held for 7 min, then to 280 °C at 20 °C min⁻¹, and held for 15 min. The injection volume was 1 µL each time in a splitless mode, and solvent delay was 5 min to prevent filament damage. The carrier gas was helium at a flow rate of 1.2 mL min⁻¹.”

6. Soil properties: what is about the organic content? Cation exchange capacities (CEC), humic and fulvic acid. They much also be degraded by this treatment?? What happens here?? Corg, how much organic carbon is actually left and in which form. This is crucial!

Response: We appreciate the reviewer’s suggestion on the soil property test. We added more tests to measure the cation exchange capacities (CEC) and organic content in the raw soil and REM soil. We discuss each aspect individually below:

(1) CEC: Soil CEC was measured by saturating the exchangeable sites with Na⁺ followed by substitution with NH₄⁺. Afterwards, the CEC can be calculated from the Na⁺ content in the final extractant. We compared the CEC of raw soil, REM soil, and calcined soil, as shown in Table R4. The REM soil has a comparable CEC content with raw soil, indicating a good

preservation of CEC after REM treatment. In contrast, the CEC of calcined soil decreased by ~73%, compared with that of raw soil, which may be ascribed to its enhanced soil particle size (Fig. 4d) and decreased surface area (Supplementary Fig. 44).

Table R4. Soil CEC contents

Soil types	CEC (cmol kg ⁻¹)
Raw soil	15.25 ± 0.81
REM soil	15.45 ± 1.04
Calcined soil	4.08 ± 0.37

(This table has been included in SI as Supplementary Table 6)

We have revised the manuscript accordingly, on P14,

“REM soil exhibits a pH of 7.58 that is slightly higher than that of raw soil (pH = 7.19) and the CEC is 15.45 cmol kg⁻¹, which is comparable to that of raw soil (15.25 cmol kg⁻¹, Supplementary Table 6).”

We have added CEC test procedures in the Method section of the revised manuscript accordingly on P29-30,

“

1.0 g of soil sample was dispersed into 10 mL of 1 M sodium acetate solution in an ultrasonic bath (Cole-Parmer Ultrasonic Cleaner) for 15 min to saturate soil exchange sites with Na⁺. Then, the sample was washed three times with ethanol (Decon’s Pure 200 Proof, Decon Labs Inc.) to remove the excess Na⁺. Afterwards, the sample was dispersed into 10 mL of 1 M ammonium acetate in an ultrasonic bath for another 15 min to replace Na⁺ by NH₄⁺ at exchange sites⁵⁵. The sample was then centrifuged (Adams analytical centrifuge, 60 rpm, 5 min), followed by filtration using a PES membrane (0.22 μm, Millipore-Sigma) to remove any undissolved particles. The filtrate was diluted to the appropriate concentration using 2 wt% HNO₃ within the ICP calibration curve range. The Na⁺ concentration was measured by inductively coupled plasma mass spectrometry (ICP-MS) using a Perkin Elmer Nexion 300 ICP-MS system, with sodium standard solution for ICP (1 g L⁻¹, Millipore Sigma) as the standard. Finally, CEC was calculated by equation (4),

$$CEC = \frac{c(\text{Na}^+) \times D \times V}{M(\text{Na}^+) \times m(\text{soil})} \quad (4)$$

where $c(\text{Na}^+)$ is the concentration of sodium measured by ICP-MS, D is the dilution factor, V is the volume of ammonium acetate solution to extract Na⁺ ($V = 10$ mL), $M(\text{Na}^+)$ is mole mass of sodium ($M(\text{Na}^+) = 23$ g mol⁻¹), $m(\text{soil})$ is the mass of soil sample used for CEC test ($m(\text{soil}) = 1.0$ g).”

”

Reference:

55. Carter, M. R. & Gregorich, E. G. *Soil sampling and methods of analysis*. (CRC press, 2007).

(2) Soil organic content: First, we tested soil carbon content by elemental combustion method (Supplementary Fig. 47). The carbon content in raw soil (~3.7 wt%) comes from the soil organic compounds, while the carbon content in REM soil (~4.3 wt%) can be mainly ascribed to the existence of small amounts of residual biochar. To further quantify the content of total dissolved organic compounds in soil, we extracted the soil organics using a mixture of NaOH and Na₄P₂O₇ solution and titrated it by the Walkley–Black method to get the content of total dissolved organic compounds. Considering humic acid is insoluble in acid, we separated the humic acid in the extractant by adding H₂SO₄, redissolving it using NaOH and then quantifying it using a similar titration method. Thus, the fulvic acid content can be calculated by subtracting humic acid content from the total dissolved organic compound content (*Org. Geochem.*, 2007, 38, 1).

Compared to the organic contents in raw soil, both humic and fulvic acid contents in REM soil decreased by >99 wt% (Fig. R25), indicating that most soil organics decomposed during REM process. However, for real-world use of REM soil in the ecosystem, these organics can be regenerated after introducing microorganisms to decompose plants or animals.

Fig. R25. Organic content in raw soil and REM soil. The error bars denote standard deviations, where $N = 3$.

(This figure has been added into the SI as Supplementary Fig. 48)

We have revised the manuscript accordingly, on P14,

“The contents of extractable organic compounds, including humic acid and fulvic acid, were quantified by the Walkley–Black method⁴⁵, where <1 wt% of these compounds remained in the REM soil, indicating the decomposition of these compounds during REM process (Supplementary Fig. 48).”

Reference

45. Baglieri, A., Ioppolo, A., Nègre, M. & Gennari, M. A method for isolating soil organic matter after the extraction of humic and fulvic acids. *Org. Geochem.* 38, 140-150 (2007).

We have added the procedures of soil organic content test in the Method section of the revised manuscript accordingly, on P30-31,

“1.0 g of soil sample was dispersed into 10 mL of extractant ($V_1 = 10$ mL), composed of 0.5 M NaOH and 0.5 M $\text{Na}_4\text{P}_2\text{O}_7$. The mixture was shaken on a horizontal shaker (Burrell Scientific Wrist Action, Model 75) for 1 h, and then heated at 95 °C for 30 min, followed by centrifuging (Adams Analytical Centrifuge, 60 rpm) for 2 min. After filtrating through a polyethersulfone (PES) membrane (0.22 μm , Millipore-Sigma), we obtained solution 1. For the total organic content test, 1 mL of solution 1 ($V_2 = 1$ mL) was mixed with 5 mL of 0.4 M $\text{K}_2\text{Cr}_2\text{O}_7$ and 15 mL of 2 M H_2SO_4 , and then heated at 95 °C for 30 min to oxidize the organic compounds in the extractant. After cooling to room temperature, the solution was mixed with 78.9 mL ultrapure water (Millipore Sigma, ACS reagent for ultratrace analysis) and 0.1 mL of phenanthroline indicator (1.5 wt% phenanthroline and 1 wt% $(\text{NH}_4)_2\text{Fe}(\text{SO}_4)_2$), which was denoted as solution 2. Afterwards, 0.1 M $(\text{NH}_4)_2\text{Fe}(\text{SO}_4)_2$ was gradually added to solution 2 until the solution's color changed from orange to green and finally to brick red. The consumed volume of $(\text{NH}_4)_2\text{Fe}(\text{SO}_4)_2$ solution was recorded as V_3 . For the comparison, 0.1 M $(\text{NH}_4)_2\text{Fe}(\text{SO}_4)_2$ was gradually added to a solution with 5 mL of 0.1 M $\text{K}_2\text{Cr}_2\text{O}_7$, 15 mL of 2 M H_2SO_4 , 74.9 mL of ultrapure water (Millipore Sigma, ACS reagent for ultratrace analysis) and 0.1 mL of phenanthroline indicator. The consumed volume of $(\text{NH}_4)_2\text{Fe}(\text{SO}_4)_2$ solution was recorded as V_0 when the solution color changed to green. Therefore, the total organic mass content (c_{org} , with the unit of g kg^{-1}) can be calculated from equation (5):

$$c_{\text{org}} = \frac{0.003 \times (V_0 - V_3) \times c(\text{Fe}^{2+}) \times r_o \times r_c}{m} \times \frac{V_1}{V_2} \times 1000 \quad (5)$$

Where $c(\text{Fe}^{2+})$ is the mole concentration of $(\text{NH}_4)_2\text{Fe}(\text{SO}_4)_2$ ($c(\text{Fe}^{2+}) = 0.1$ M), m is the soil mass ($m = 1$ g), r_o and r_c is the oxidation factor and the conversion factor from organic carbon to organic compound ($r_o = 1.1$ and $r_c = 1.724$).

Considering the insolubility of humic acid in acid solution, 2 M H_2SO_4 was added to 5 mL of solution 1 ($V_5 = 5$ mL) until the pH reached 1 and it was then left for 30 min to separate humic acid from the soil extractant. After filtering using a sand core funnel (class F), the filter residue was washed by 0.05 M H_2SO_4 for 5 times. Afterwards, the residue was dissolved by 50 mL 1 wt% NaOH and then diluted to 100 mL using ultrapure water (Millipore Sigma, ACS reagent for ultratrace analysis), which is denoted as Solution 3 ($V_6 = 100$ mL). Similarly, 5 mL of 0.4 M $\text{K}_2\text{Cr}_2\text{O}_7$ and 15 mL of 2 M H_2SO_4 were used to oxidize 5 mL Solution 3 ($V_7 = 5$ mL) at 95 °C for 30 min and the residue $\text{K}_2\text{Cr}_2\text{O}_7$ in solution was titrated by 0.1 M $(\text{NH}_4)_2\text{Fe}(\text{SO}_4)_2$ using the phenanthroline indicator. The consumed volume of $(\text{NH}_4)_2\text{Fe}(\text{SO}_4)_2$

solution was recorded as V_8 . The humic acid mass content (c_{humic} , with the unit of g kg^{-1}) can be calculated from equation (6):

$$c_{\text{humic}} = \frac{0.003 \times (V_0 - V_8) \times c(\text{Fe}^{2+}) \times r_o \times r_c}{m} \times \frac{V_1}{V_5} \times \frac{V_6}{V_7} \times 1000 \quad (6)$$

The fulvic acid content (c_{fulvic} , with the unit of g kg^{-1}) can be thus calculated by:

$$c_{\text{fulvic}} = c_{\text{org}} - c_{\text{humic}} \quad (7)$$

REVIEWERS' COMMENTS

Reviewer #1 (Remarks to the Author):

My comments have been properly addressed

Reviewer #2 (Remarks to the Author):

The authors have correctly addressed the comments. I recommend this paper for publication.

Reviewer #3 (Remarks to the Author):

Dear Editor

the authors made a really good effort to address my comments in the rebuttal and the amendments in the manuscript are appropriate.

I would like only to point out that they might want to discuss the new results more. It seems that CEC but more so the soluble DOC as HA and FA alters after treatment significantly. What does it mean in terms of the use of the soil?

Otherwise I am happy with the revision and I would recommend to publish this paper.

Reviewer #1 (Remarks to the Author):

Comments:

My comments have been properly addressed

Response: We appreciate the reviewer for the kind review of our work.

Reviewer #2 (Remarks to the Author):

Comments:

The authors have correctly addressed the comments. I recommend this paper for publication.

Response: We appreciate the reviewer for the kind review of our work.

Reviewer #3 (Remarks to the Author):

Comments:

The authors made a really good effort to address my comments in the rebuttal and the amendments in the manuscript are appropriate. I would like only to point out that they might want to discuss the new results more. It seems that CEC but more so the soluble DOC as HA and FA alters after treatment significantly. What does it mean in terms of the use of the soil?

Otherwise I am happy with the revision and I would recommend to publish this paper.

Response: We appreciate the reviewer for the kind review of our work. REM soil has a comparable CEC (15.45 cmol kg⁻¹) to that of raw soil (15.25 cmol kg⁻¹, Supplementary Table 6). However, as the reviewer mentioned, >99 wt% of extractable organic compounds, including humic acid and fulvic acid, were decomposed during the REM process (Supplementary Fig. 48). We agree that higher contents of these organic compounds in a certain range can indeed facilitate soil health and plant growth. But for the practical use of

REM soil with low contents of organic compounds, microorganisms/invertebrates can quickly colonize the soil and provide the soil with organic compounds by decomposing plant remaining and/or animal carcasses and feces (*Biogeochemistry*, 1990, 11, 213-233; *Eur. J. Soil Sci.* 1999, 50, 567-578). It indicates that soil with low organic content is still viable.

**We have added the discussion in the revised manuscript accordingly, on P14,
“For the practical use, the organic contents in REM soil can be easily recovered by introducing microorganisms to decompose plant/animal remaining.”**